# Energy-Based Learning for Cooperative Games, with Applications to Valuation Problems in Machine Learning

**Yatao Bian[1]***, **Yu Rong[1], Tingyang Xu[1], Jiaxiang Wu[1], Andreas Krause[2], Junzhou Huang[1]**
[1] Tencent AI Lab          [2] ETH Zürich

## Abstract

Valuation problems, such as feature interpretation, data valuation and model valuation for ensembles, become increasingly more important in many machine learning applications. Such problems are commonly addressed via well-known game-theoretic criteria, such as the Shapley value or Banzhaf value. In this work, we present a novel energy-based treatment for cooperative games, with a theoretical justification via the maximum entropy principle. Surprisingly, through mean-field variational inference in the energy-based model, we recover classical game-theoretic valuation criteria by conducting *one-step* of fixed point iteration for maximizing the ELBO objective. This observation also further supports existing criteria, as they can be seen as attempting to decouple the correlations among players. By running the fixed point iteration for *multiple* steps, we achieve a trajectory of the variational valuations, among which we define the valuation with the best conceivable decoupling error as the *Variational Index*. We prove that under uniform initialization, these variational valuations all satisfy a set of game-theoretic axioms. We empirically demonstrate that the proposed variational valuations enjoy lower decoupling error and better valuation performance on certain synthetic and real-world valuation problems.[1]

## 1 Introduction

Valuation problems are becoming increasingly more significant in various machine learning applications, ranging from feature interpretation (Lundberg and Lee, 2017), data valuation (Ghorbani and Zou, 2019) to model valuation for ensembles (Rozemberczki and Sarkar, 2021). They are often formulated as a player valuation problem in cooperative games. A cooperative game $(N, F(S))$ consists of a grand coalition $N = \{1, ..., n\}$ of $n$ players and a value function (a.k.a. characteristic function) $F(S) : 2^N \to \mathbb{R}$ describing the collective payoff of a coalition/cooperation $S$. A fundamental problem in cooperative game theory is to assign an importance vector (i.e., solution concept) $\phi(F) \in \mathbb{R}^n$ to $n$ players.

In this paper, we explore a *probabilistic treatment* of cooperative games $(N, F(S))$. Such a treatment makes it possible to conduct learning and inference in a unified manner, and will yield connections with classical valuation criteria. Concretely, we seek a probability distribution over coalitions $p(\mathbf{S} = S)$[2], measuring the odds that a specific coalition

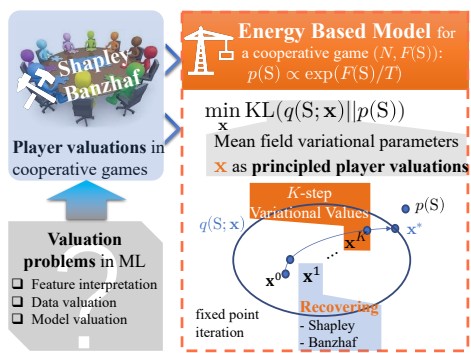

Figure 1: An energy based treatment of cooperative games leads to a series of new player valuations: $K$-step variational values, satisfying basic valuation axioms: null player, marginalism & symmetry.

---

*Correspondence to: Yatao Bian <yatao.bian@gmail.com>

[1]Project page & code: https://valuationgame.github.io

[2]Note that distributions over subsets of $N$ are equivalent to distributions of $|N| = n$ binary random variables $X_1, ..., X_n \in \{0, 1\}$: We use $X_i$ as the indicator function of the event $i \in S$, or $X_i = [i \in S]$. With slight abuse of notation, we use $\mathbf{S}$ as a random variable represented as sets and often abbreviate $p(\mathbf{S} = S)$ as $p(S)$.

$S$ happens. Generally, we consider distributions where the probability of a coalition $p(S)$ grows monotonically with the payoff $F(S)$.

Among all the possible probability mass functions (pmfs), how should we construct the proper $p(S)$? We advocate to choose the pmf with the maximum entropy $\mathbb{H}(p)$. This principle makes sense since maximizing the entropy minimizes the amount of prior information built into the distribution. In other words, it amounts to assuming nothing about what is unknown, i.e., choosing the most "uniform" distribution. Now finding a proper $p(S)$ becomes the following constrained optimization problem: suppose each coalition $S$ is associated with a payoff $F(S)$ with probability $p(S)$. We would like to maximize the entropy $\mathbb{H}(p) = -\sum_{S \subseteq N} p(S) \log p(S)$, subject to the constraints that $\sum_S p(S) = 1, p(S) \geq 0$ and $\sum_S p(S)F(S) = \mu$ (i.e., the average payoff is known as $\mu$). Solving this optimization problem (derivation in Appendix A), we reach the *maximum entropy distribution*:

$$p(S) = \frac{\exp(F(S)/T)}{\mathbf{Z}}, \quad \mathbf{Z} := \sum_{S' \subseteq N} \exp(F(S')/T), \tag{1}$$

where $T > 0$ is the temperature. This is an energy-based model (EBM, cf. LeCun et al., 2006) with $-F(S)$ as the energy function.

The above energy-based treatment admits two benefits: i) Where supervision is available, it enables learning of value functions $F(S)$ through efficient training techniques for energy-based learning, such as noise contrastive estimation (Gutmann and Hyvärinen, 2010) and score matching (Hyvärinen, 2005). ii) Approximate inference techniques such as variational inference or sampling can be adopted to solve the valuation problem. Specifically, it enables to perform mean-field variational inference where parameters of the inferred surrogate distribution can be used as principled player valuations.

Below, we explore mean-field variational inference for the energy-based formulation (Fig. 1). Perhaps surprisingly, by conducting only *one-step* fixed point iteration for maximizing the mean-field (ELBO) objective, we recover classical valuation criteria, such as the Shapley value (Shapley, 1953) and the Banzhaf value (Penrose, 1946; Banzhaf III, 1964). This observation also further supports existing criteria, motivating them as decoupling the correlations among players via the mean-field approach. By running the fixed point iteration for *multiple* steps, we achieve a trajectory of valuations, among which we define the valuation with the best conceivable decoupling error as the *Variational Index*. Our major contributions can be summarized as below:

i) We present a theoretically justified energy-based treatment for cooperative games. Through mean field inference, we provide a unified perspective on popular game-theoretic criteria. This provides an alternative motivation of existing criteria via a *decoupling* perspective, i.e., decoupling correlations among $n$ players through the mean-field approach. ii) In pursuit of better decoupling performance, we propose to run fixed point iteration for *multiple* steps, which generates a trajectory of valuations. Under uniform initializations, they all satisfy a set of game-theoretic axioms, which are required for being suitable valuation criteria. We define the valuation with the best conceivable decoupling error as the Variational Index. iii) Synthetic and real-world experiments demonstrate intriguing properties of the proposed Variational Index, including lower decoupling error and better valuation performance.

## 2 PRELIMINARIES AND BACKGROUND

**Notation.** We assume $\mathbf{e}_i \in \mathbb{R}^n$ being the standard $i^{\text{th}}$ basis vector and use boldface letters $\mathbf{x} \in \mathbb{R}^N$ and $\mathbf{x} \in \mathbb{R}^n$ interchangebly to indicate an $n$-dimensional vector, where $x_i$ is the $i^{\text{th}}$ entry of $\mathbf{x}$. By default, $f(\cdot)$ is used to denote a continuous function, and $F(\cdot)$ to represent a set function. For a differentiable function $f(\cdot)$, $\nabla f(\cdot)$ denotes its gradient. $\mathbf{x}|x_i \leftarrow k$ is the operation of setting the $i^{\text{th}}$ element of $\mathbf{x}$ to $k$, while keeping all other elements unchanged, i.e., $\mathbf{x}|x_i \leftarrow k = \mathbf{x} - x_i\mathbf{e}_i + k\mathbf{e}_i$. For two sets $S$ and $T$, $S + T$ and $S - T$ represent set union and set difference, respectively. $|S|$ is the cardinality of $S$. $i$ is used to denote the singleton $\{i\}$ with a bit abuse of notation.

**Existing valuation criteria.** Various valuation criteria have been proposed from the area of cooperative games, amongst them the most famous ones are the Shapley value (Shapley, 1953) and the Banzhaf value, which is extended from the Banzhaf power index (Penrose, 1946; Banzhaf III, 1964). For the Shapley value, the importance assigned to player $i$ is:

$$\mathbf{Sh}_i = \sum_{S \subseteq N-i} [F(S+i) - F(S)] \frac{|S|!(n-|S|-1)!}{n!}. \tag{2}$$

One can see that it gives less weight to $n/2$-sized coalitions. The Banzhaf value assigns the following importance to player $i$:

$$\text{Ba}_i = \sum_{S \subseteq N-i}[F(S+i) - F(S)]\frac{1}{2^{n-1}}, \tag{3}$$

which uses uniform weights for all the coalitions. See Greco et al. (2015) for a comparison of them.

**Valuation problems in machine learning.** Currently, most classes of valuation problems (Lundberg and Lee, 2017; Ghorbani and Zou, 2019; Sim et al., 2020; Rozemberczki and Sarkar, 2021) use Shapley value as the valuation criterion. Along with the rapid progress of model interpretation in the past decades (Zeiler and Fergus, 2014; Ribeiro et al., 2016; Lundberg and Lee, 2017; Sundararajan et al., 2017; Petsiuk et al., 2018; Wang et al., 2021a), attribution-based interpretation aims to assign importance to the features for a specific data instance $(\mathbf{x} \in \mathbb{R}^N, y)$ given a black-box model $\mathcal{M}$. Here each feature maps to a player in the game $(N, F(S))$, and the value function $F(S)$ is usually the model response, such as the predicted probability for classification problems, when feeding a subset $S$ of features to $\mathcal{M}$. The data valuation problem (Ghorbani and Zou, 2019) tries to assign values to the samples in the training dataset $N = \{(\mathbf{x}_i, y_i)\}_1^n$ for general supervised machine learning: one training sample corresponds to one player, and the value function $F(S)$ indicates the predictor performance on some test dataset given access to only a subset of the training samples in $S$. Model valuation in ensembles (Rozemberczki and Sarkar, 2021) measures importance of individual models in an ensemble in order to correctly label data points from a dataset, where each pre-trained model maps to a player and the value function measures the predictive performance of subsets of models.

## 3    RELATED WORK

**Energy-based modeling.** Energy based learning (LeCun et al., 2006) is a classical learning framework that uses an energy function $E(\mathbf{x})$ to measure the quality of a data point $\mathbf{x}$. Energy based models have been applied to different domains, such as data generation (Deng et al., 2020), out-of-distribution detection (Liu et al., 2020), reinforcement learning (Haarnoja et al., 2017), memory modeling (Bartunov et al., 2019), discriminative learning (Grathwohl et al., 2019; Gustafsson et al., 2020) and biologically-plausible training (Scellier and Bengio, 2017). Energy based learning admits principled training methods, such as contrastive divergence (Hinton, 2002), noise contrastive estimation (Gutmann and Hyvärinen, 2010) and score matching (Hyvärinen, 2005). For approximate inference, sampling based approaches are mainly MCMC-style algorithms, such as stochastic gradient Langevin dynamics (Welling and Teh, 2011). For a wide class of EBMs with submodular or supermodular energies (Djolonga and Krause, 2014), there exist provable mean field inference algorithms with constant factor approximation guarantees (Bian et al., 2019; Sahin et al., 2020; Bian et al., 2020).

**Shapley values in machine learning.** Shapley values have been extensively used for valuation problems in machine learning, including attribution-based interpretation (Lipovetsky and Conklin, 2001; Cohen et al., 2007; Strumbelj and Kononenko, 2010; Owen, 2014; Datta et al., 2016; Lundberg and Lee, 2017; Chen et al., 2018; Lundberg et al., 2018; Kumar et al., 2020; Williamson and Feng, 2020; Covert et al., 2020b; Wang et al., 2021b), data valuation (Ghorbani and Zou, 2019; Jia et al., 2019b;a; Wang et al., 2020; Fan et al., 2021), collaborative machine learning (Sim et al., 2020) and recently, model valuation in ensembles (Rozemberczki and Sarkar, 2021). For a detailed overview of papers using Shapley values for feature interpretation, please see Covert et al. (2020a) and the references therein. To alleviate the exponential computational cost of exact evaluation, various methods have been proposed to approximate Shapley values in polynomial time (Ancona et al., 2017; 2019). Owen (1972) proposes the multilinear extension purely as a representation of cooperative games and Okhrati and Lipani (2021) use it to develop sampling algorithms for Shapley values.

## 4    VALUATION FOR COOPERATIVE GAMES: A DECOUPLING PERSPECTIVE

In the introduction, we have asserted that under the setting of cooperative games, the Boltzmann distribution (see Eq. (1)) achieves the maximum entropy among all of the pmf functionals. One can naturally view the importance assignment problem of cooperative games as a *decoupling problem*: The $n$ players in a game $(N, F(S))$ might be arbitrarily correlated in a very complicated manner. However, in order to assign each of them an *individual* importance value, we have to decouple their interactions, which can be viewed as a way to simplify their correlations.

We therefore consider a surrogate distribution $q(S; \mathbf{x})$ governed by parameters in $\mathbf{x}$. $q$ has to be simple, given our intention to decouple the correlations among the $n$ players. A natural choice is to restrain $q(S; \mathbf{x})$ to be fully factorizable, which leads to a mean-field approximation of $p(S)$. The simplest form of $q(S; \mathbf{x})$ would be a $n$ independent Bernoulli distribution, i.e., $q(S; \mathbf{x}) := \prod_{i \in S} x_i \prod_{j \notin S} (1 - x_j), \mathbf{x} \in [0, 1]^n$. Given a divergence measure $D(\cdot \| \cdot)$ for probability distributions, we can define the *best conceivable decoupling error* to be the divergence between $p$ and the best possible $q$.

**Definition 1** (Best Conceivable Decoupling Error). *Considering a cooperative game $(N, F(S))$, and given a divergence measure $D(\cdot \| \cdot)$ for probability distributions, the* decoupling error *is defined as the divergence between $q$ and $p$: $D(q\|p)$, and the best conceivable decoupling error is defined as the divergence between the best possible $q$ and $p$:*

$$D^* := \min_q D(q\|p). \tag{4}$$

Note that the *best conceivable decoupling error* $D^*$ is closely related to the intrinsic coupling amongst $n$ players: if all the players are already independent with each other, then $D^*$ could be zero.

## 4.1 MEAN FIELD OBJECTIVE FOR EBMS

If we consider the *decoupling error* $D(q\|p)$ to be the Kullback-Leibler divergence between $q$ and $p$, then we recover the mean field approach[3]. Given the EBM formulation in Eq. (1), the classical mean-field inference approach aims to approximate $p(S)$ by a fully factorized product distribution $q(S; \mathbf{x}) := \prod_{i \in S} x_i \prod_{j \notin S} (1 - x_j), \mathbf{x} \in [0, 1]^n$, by minimizing the distance measured w.r.t. the Kullback-Leibler divergence between $q$ and $p$. Since $\mathbb{KL}(q\|p)$ is non-negative, we have:

$$0 \leq \mathbb{KL}(q\|p) = \sum_{S \subseteq N} q(S; \mathbf{x}) \log \frac{q(S; \mathbf{x})}{p(S)}$$

$$= -\mathbb{E}_{q(S;\mathbf{x})}[\log p(S)] - \mathbb{H}(q(S; \mathbf{x})) \tag{5}$$

$$= -\mathbb{E}_{q(S;\mathbf{x})}\left[\frac{F(S)}{T} - \log \mathbf{Z}\right] - \mathbb{H}(q(S; \mathbf{x})) \tag{6}$$

$$= -\sum_{S \subseteq N} \frac{F(S)}{T} \prod_{i \in S} x_i \prod_{j \notin S} (1 - x_j) + \sum_{i=1}^{n} [x_i \log x_i + (1 - x_i) \log(1 - x_i)] + \log \mathbf{Z}. \tag{7}$$

In Eq. (6) we plug in the EBM formulation that $\log p(S) = \frac{F(S)}{T} - \log \mathbf{Z}$. Then one can get

$$\log \mathbf{Z} \geq \sum_{S \subseteq N} \frac{F(S)}{T} \prod_{i \in S} x_i \prod_{j \notin S} (1 - x_j) - \sum_{i=1}^{n} [x_i \log x_i + (1 - x_i) \log(1 - x_i)]$$

$$= \frac{f_{\text{mt}}^F(\mathbf{x})}{T} + \mathbb{H}(q(S; \mathbf{x})) := (\text{ELBO}) \tag{8}$$

where $\mathbb{H}(\cdot)$ is the entropy, ELBO stands for the *evidence lower bound*, and

$$f_{\text{mt}}^F(\mathbf{x}) := \sum_{S \subseteq N} F(S) \prod_{i \in S} x_i \prod_{j \notin S} (1 - x_j), \mathbf{x} \in [0, 1]^n, \tag{9}$$

is the *multilinear extension* of $F(S)$ (Owen, 1972; Calinescu et al., 2007). Note that the multilinear extension plays a central role in modern combinatorial optimizaiton techniques (Feige et al., 2011), especially for guaranteed submodular maximization problems (Krause and Golovin, 2014).

Maximizing (ELBO) in Eq. (8) amounts to minimizing the Kullback-Leibler divergence between $q$ and $p$. If one solves this optimization problem to optimality, one can obtain the $q(S; \mathbf{x}^*)$ with the best conceivable decoupling error. Here $x_i^*$ describes the odds that player $i$ shall participate in the game, so it can be naturally used to define the importance score of each player.

**Definition 2** (Variational Index of Cooperative Games). *Consider a cooperative game $(N, F(S))$ and its mean field approximation. Let $\mathbf{x}^*$ be the variational marginals with the best conceivable decoupling error, we define $\mathbf{s}^* := T\sigma^{-1}(\mathbf{x}^*)$ to be the variational index of the game. Formally,*

$$\mathbf{x}^* = \arg\min_{\mathbf{x}} \mathbb{KL}(q(S; \mathbf{x})\|p(S)), \tag{10}$$

*where $\mathbf{x}^*$ can be obtained by maximizing the ELBO objective in Eq. (8), and $\sigma^{-1}(\cdot)$ is the inverse of the sigmoid function, i.e. $\sigma^{-1}(x) = \log \frac{x}{1-x}$. For a vector it is applied element-wise.*

---

[3]Notably, one could also apply the reverse KL divergence $\mathbb{KL}(p\|q)$, which would lead to an expectation propagation (Minka, 2001) treatment of cooperative games.

### 4.2 ALGORITHMS FOR CALCULATING THE VARIATIONAL INDEX

**Equilibrium condition.** For coordinate $i$, the partial derivative of the multilinear extension is $\nabla_i f_{\mathrm{mt}}^F(\mathbf{x})$, and for the entropy term it is $\nabla_i \mathbb{H}(q(S; \mathbf{x})) = \log \frac{1-x_i}{x_i}$. By setting the partial derivative of ELBO in Eq. (8) to be 0, we have the equilibrium condition:

$$x_i^* = \sigma(\nabla_i f_{\mathrm{mt}}^F(\mathbf{x}^*)/T) = \left(1 + \exp(-\nabla_i f_{\mathrm{mt}}^F(\mathbf{x}^*)/T)\right)^{-1}, \quad \forall i \in N, \tag{11}$$

where $\sigma$ is the sigmoid function. This equilibrium condition implies that one cannot change the value assigned to any player in order to further improve the overall decoupling performance. It also implies the fixed point iteration $x_i \leftarrow \sigma(\nabla_i f_{\mathrm{mt}}^F(\mathbf{x})/T)$. When updating each coordinate sequentially, we recover the classic naive mean field algorithm as shown in Appendix C.

Instead, here we suggest to use the full-gradient method shown in Alg. 1 for maximizing the ELBO objective. As we will see later, the resultant valuations satisfy certain game-theoretic axioms. It needs an initial marginal vector $\mathbf{x}^0 \in [0,1]^n$ and the number of epochs $K$. After $K$ steps of fixed point iteration, it returns the estimated marginal $\mathbf{x}^K$.

---

**Algorithm 1:** Mean Field Inference with Full Gradient: MFI($\mathbf{x}; K$)

**Input:** A cooperative game $(N, F(S))$ with $n$ players. Initial marginals $\mathbf{x}^0 \leftarrow \mathbf{x} \in [0,1]^n$.
   #epochs $K$.
**Output:** Marginals after $K$ steps of iteration: $\mathbf{x}^K$

1  **for** $k = 1 \rightarrow K$ **do**
2  $\quad \lfloor \quad \mathbf{x}^k \leftarrow \sigma(\nabla f_{\mathrm{mt}}^F(\mathbf{x}^{k-1})/T) = \left(1 + \exp(-\nabla f_{\mathrm{mt}}^F(\mathbf{x}^{k-1})/T)\right)^{-1}$ ;

---

In case Alg. 1 solves the optimization problem to optimality, we obtain the Variational Index. However, maximizing ELBO is in general a non-convex/non-concave problem, and hence one can only ensure reaching a stationary solution. Below, when we say Variational Index, we therefore refer to its approximation obtained via Alg. 1 by default. Meanwhile, the MFI($\mathbf{x}; K$) subroutine also defines a series of marginals, which enjoy interesting properties as we show in the next part. So we define variational valuations through intermediate solutions of MFI($\mathbf{x}; K$).

**Definition 3** ($K$-Step Variational Values). *Considering a cooperative game $(N, F(S))$ and its mean field approximation by Alg. 1, we define the $K$-Step Variational Values initialized at $\mathbf{x}$ as:*

$$T\sigma^{-1}(MFI(\mathbf{x}; K)), \tag{12}$$

*where $\sigma^{-1}()$ is the inverse of the sigmoid function ($\sigma^{-1}(x) = \log \frac{x}{1-x}$).*

Notice when running more steps, the $K$-Step variational value will be more close to the Variational Index. The gradient $\nabla f_{\mathrm{mt}}^F(\mathbf{x})$ itself is defined with respect to an exponential sum via the multilinear extension. Next we show how it can be approximated via principled sampling methods.

**Sampling methods for estimating the partial derivative.** The partial derivative follows,

$$\nabla_i f_{\mathrm{mt}}^F(\mathbf{x}) = \mathbb{E}_{q(S;(\mathbf{x}|x_i \leftarrow 1))}[F(S)] - \mathbb{E}_{q(S;(\mathbf{x}|x_i \leftarrow 0))}[F(S)] \tag{13}$$

$$= f_{\mathrm{mt}}^F(\mathbf{x}|x_i \leftarrow 1) - f_{\mathrm{mt}}^F(\mathbf{x}|x_i \leftarrow 0)$$

$$= \sum_{S \subseteq N, S \ni i} F(S) \prod_{j \in S-i} x_j \prod_{j' \notin S} (1 - x_{j'}) \; - \sum_{S \subseteq N-i} F(S) \prod_{j \in S} x_j \prod_{j' \notin S, j' \neq i} (1 - x_{j'})$$

$$= \sum_{S \subseteq N-i} [F(S+i) - F(S)] \prod_{j \in S} x_j \prod_{j' \in N-S-i} (1 - x_{j'})$$

$$= \sum_{S \subseteq N-i} [F(S+i) - F(S)] q(S; (\mathbf{x}|x_i \leftarrow 0)) = \mathbb{E}_{S \sim q(S;(\mathbf{x}|x_i \leftarrow 0))} [F(S+i) - F(S)].$$

All of the variational criteria are based on the calculation of the partial derivative $\nabla_i f_{\mathrm{mt}}^F(\mathbf{x})$, which can be approximated by Monte Carlo sampling since $\nabla_i f_{\mathrm{mt}}^F(\mathbf{x}) = \mathbb{E}_{S \sim q(S;(\mathbf{x}|x_i \leftarrow 0))} [F(S+i) - F(S)]$: we first sample $m$ coalitions $S_k, k = 1, ..., m$ from the surrogate distribution $q(S; (\mathbf{x}|x_i \leftarrow 0))$, then approximate the expectation by the average $\frac{1}{m} \sum_{k=1}^{m} [F(S_k + i) - F(S_k)]$. According to the

Chernoff-Hoeffding bound (Hoeffding, 1963), the approximation will be arbitrarily close to the true value with increasingly more samples: With probability at least $1 - \exp(-m\epsilon^2/2)$, it holds that $|\frac{1}{m}\sum_{k=1}^{m}[F(S_k + i) - F(S_k)] - \nabla_i f_{\mathrm{mt}}^F(\mathbf{x})| \leq \epsilon \max_S |F(S + i) - F(S)|$, for all $\epsilon > 0$.

**Roles of the initializer $\mathbf{x}^0$ and the temperature $T$.** This can be understood in the following respects: 1) The initializer $\mathbf{x}^0$ represents the initial credit assignments to the $n$ players, so it denotes the prior knowledge/initial belief of the contributions of the players; 2) If one just runs Alg. 1 for one step, $\mathbf{x}^0$ matters greatly to the output. However, if one runs Alg. 1 for many steps, $\mathbf{x}^k$ will converge to the stationary points of the ELBO objective. Empirically, it takes around 5∼10 steps to converge. The temperature $T$ controls the "spreading" of importance assigned to the players: A higher $T$ leads to flatter assignments, and a lower $T$ leads to more concentrated assignments.

**Computational efficiency of calculating Variational Index and variational values.** Alg. 1 calculates the $K$-step variational values, 1-step variational value has the same computational cost as that of Banzhaf value and of the integrand of the line integration of Shapley value in Eq. (15) below, since they all need to evaluate $\nabla f_{\mathrm{mt}}^F(\mathbf{x})$. Sampling methods could help with approximating all of the three criteria when there are a large number of players. The Variational Index can be approximated by the $K$-step variational value, where the number $K$ depends on when Alg. 1 converges. One can easily show that, under the setting of maximizing ELBO, $\mathbf{x}^k$ will converge to some stationary point $\mathbf{x}^*$, based on the analysis of mean field approximation in Wainwright and Jordan (2008). We have also empirically verified the convergence rate of Alg. 1 in Sec. 5.3, and find that it converges within 5 to 10 steps. So the computational cost is roughly similar as that of Shapley value and Banzhaf value.

### 4.3 RECOVERING CLASSICAL CRITERIA

Perhaps surprisingly, it is possible to recover classical valuation criteria via the $K$-step variational values as in Def. 3. Firstly, for Banzhaf value, by comparing with Eq. (13) it reads,

$$\mathrm{Ba}_i = \sum_{S \subseteq N - i}[F(S + i) - F(S)]\frac{1}{2^{n-1}} = \nabla_i f_{\mathrm{mt}}^F(0.5 * \mathbf{1}) = T\sigma^{-1}(\mathrm{MFI}(0.5 * \mathbf{1}; 1)), \quad (14)$$

which is the 1-step variational value initialied at $0.5 * \mathbf{1}$. We can also recover the Shapley value through its connection to the multilinear extension (Owen, 1972; Grabisch et al., 2000):

$$\mathrm{Sh}_i = \int_0^1 \nabla_i f_{\mathrm{mt}}^F(x\mathbf{1})dx = \int_0^1 T\sigma^{-1}(\mathrm{MFI}(x\mathbf{1}; 1))dx, \quad (15)$$

where the integration denotes integrating the partial-derivative of the multilinear extension along the main diagonal of the unit hypercube. A self-contained proof is given in Appendix D.

These insights offer a novel, unified interpretation of the two classical valuation indices: both the Shapley value and Banzhaf value can be viewed as approximating the variational index by running *one step* of fixed point iteration for the decoupling (ELBO) objective. Specifically, for the Banzhaf value, it initializes $\mathbf{x}$ at $0.5 * \mathbf{1}$, and runs one step of fixed point iteration. For the Shapley value, it also performs a one-step fixed point approximation. However, instead of starting at a single initial point, it averages over all possible initializations through the line integration in Eq. (15).

**Relation to probabilistic values.** Probabilistic values for games (Weber, 1988; Monderer and Samet, 2002) capture a class of solution concepts, where the value of each player is given by some averaging of the player's marginal contributions to coalitions, and the weights depend on the coalitions only. According to (Monderer and Samet, 2002, Equation (3.1)), a solution $\phi$ is called a probabilistic value, if for each player $i$, there exists a probability $p^i \in \Delta(C^i)$, such that $\phi_i$ is the expected marginal contribution of $i$ w.r.t. $p^i$. Namely, $\phi_i = \sum_{S \in C^i} p^i(S)[F(S + i) - F(S)]$, where $C^i$ is the set of all subsets of $N - i$, and $\Delta(C^i)$ is the set of all probability measures on $C^i$. One can easily see that, for any fixed $\mathbf{x}$, 1-step variational value in Def. 3 is a probabilistic value with $p^i(S) = q(S; (\mathbf{x}|x_i \leftarrow 0))$, where $q(S; \mathbf{x})$ is the surrogate distribution in our EBM framework.

### 4.4 AXIOMATISATION OF $K$-STEP VARIATIONAL VALUES

Our EBM framework introduces a series of variational values controlled by $T$ and the running step number $K$. We now establish that the variational values $T\sigma^{-1}(\mathrm{MFI}(\mathbf{x}; K))$ in Def. 3 satisfy certain game-theoretic axioms (see Appendix B for definitions of five common axioms: Null player,

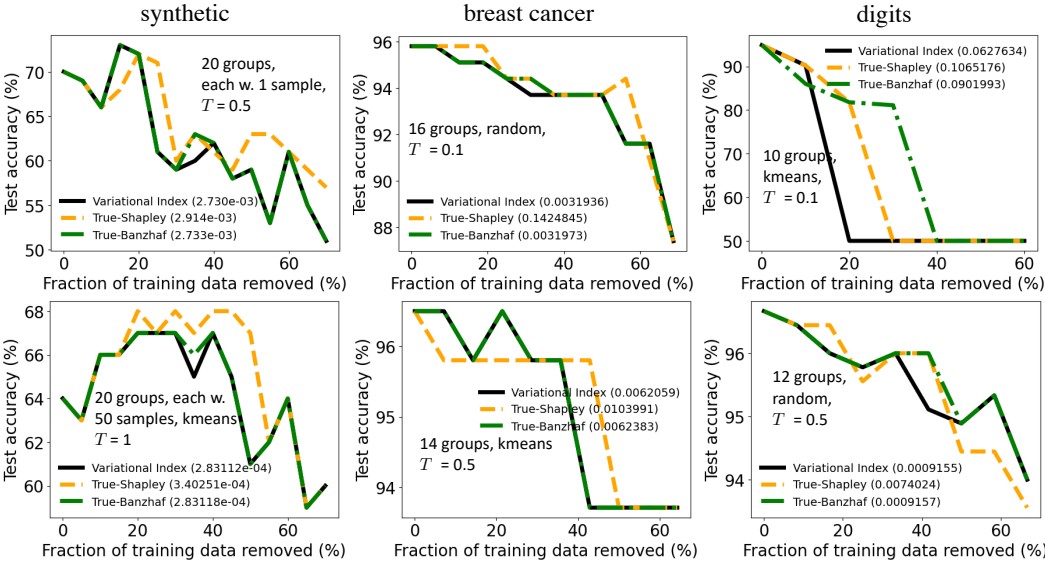

Figure 2: Data removal results. Numbers in the legend are the *decoupling* errors. Columns: 1st: synthetic data; 2nd: breast cancer data with 569 samples; 3rd: digits data with 1797 samples. Specific configurations (e.g., temperature) are put inside the figure texts.

Symmetry, Marginalism, Additivity and Efficiency). We prove that all the variational values in the trajectory satisfy three fundamental axioms: null player, marginalism and symmetry. The detailed proof is deferred to Appendix E. We expect it to be very difficult to find *equivalent* axiomatisations of the series of variational values, which we leave for future work. Meanwhile, our methods incur a decoupling and fairness tradeoff by tuning the hyperparameters $K$ and $T$.

**Theorem 1** (Axiomatisation of $K$-Step Variational Values of Def. 3). *If initialized uniformly, i.e.,* $\mathbf{x}^0 = x\mathbf{1}, x \in [0, 1]$, *all the variational values in the trajectory* $T\sigma^{-1}(MFI(\mathbf{x}; k)), k = 1, 2, 3...$ *satisfy the null player, marginalism and symmetry axioms.*

According to Theorem 1, our proposed $K$-step variational values satisfy the minimal set of axioms often associated with appropriate valuation criteria. Note that specific realizations of the $K$-step variational values can also satisfy more axioms, for example, the 1-step variational value initialized at $0.5 * \mathbf{1}$ also satisfies the additivity axiom. Furthermore, we have the following observations:

**Satisfying more axioms is not essential for valuation problems.** Notably, in cooperative game theory, one line of work is to seek for solution concepts that would satisfy more axioms. However, for valuation problems in machine learning, this is arguably not essential. For example, similar as argued by Ridaoui et al. (2018), efficiency does not make sense for certain games. We give a simple illustration in Appendix F, which further shows that whether more axioms shall be considered really depends on the specific scenario being modeled, which will be left for important future work.

## 5 EMPIRICAL STUDIES

Throughout the experiments, we are trying to understand the following: 1) Would the proposed Variational Index have lower decoupling error compared to others? 2) Could the proposed Variational Index gain benefits compared to the classical valuation criteria for valuation problems?

Since we are mainly comparing the quality of different criteria, it is necessary to rule out the influence of approximation errors when estimating their values. So we focus on small-sized problems where one can compute the exact values of these criteria in a reasonable time. Usually this requires the number of players to be no more than 25. Meanwhile, we have also conducted experiments with a larger number of players in Appendix G.5, in order to show the efficiency of sampling methods. We choose $T$ empirically from the values of 0.1, 0.2, 0.5, 1.0. We choose $K$ such that Alg. 1 would converge. Usually, it takes around 5 to 10 steps to converge. We give all players a fair start, so $\mathbf{x}^0$ was intialized to be $0.5 \times \mathbf{1}$. Code is available at https://valuationgame.github.io.

We first conduct synthetic experiments on submodular games (details defered to Appendix G.1), in order to verify the quality of solutions in terms of the true marginals $p(i \in \mathbf{S})$. One can conclude

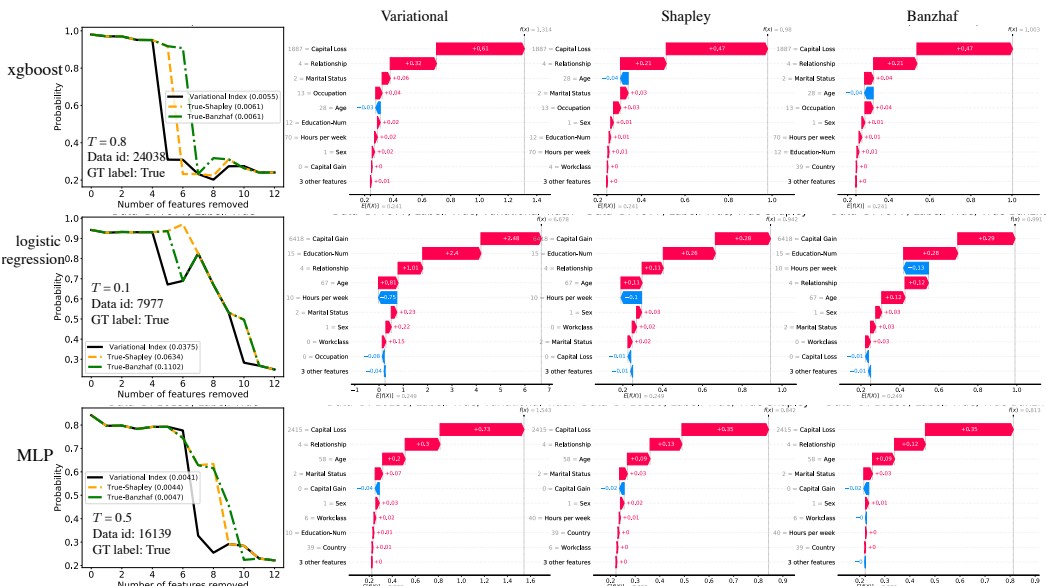

Figure 3: First column: Change of predicted probabilities when removing features. The *decoupling error* is included in the legend. Last three columns: waterfall plots of feature importance.

that Variational Index obtains better performance in terms of MSE and Spearman's rank correlation compared to the one-point solutions (Shapley value and Banzhaf value) in all experiments. More experimental results on data point and feature valuations are deferred to Appendix G.

## 5.1 EXPERIMENTS ON DATA VALUATIONS

We follow the setting of Ghorbani and Zou (2019) and reuse the code of `https://github.com/amiratag/DataShapley`. We conduct data removal: training samples are sorted according to the valuations returned by different criteria, and then samples are removed in that order to check how much the test accuracy drops. Intuitively, the best criteria would induce the fastest drop of performance. We experiment with the following datasets: a) Synthetic datasets similar as that of Ghorbani and Zou (2019); b) The breast cancer dataset, which is a binary classification dataset with 569 samples; c) The digits dataset, that is a 10-class classification dataset with 1797 samples. The above two datasets are both from UCI Machine Learning repository (`https://archive.ics.uci.edu/ml/index.php`). Specifically, we cluster data points into groups and studied two settings: 1) Grouping the samples randomly; 2) Clustering the samples with the k-means algorithm. For simplicity, we always use equal group sizes. The data point removal corresponds to singleton groups. Fig. 2 shows the results. One can observe that in certain situations the Variational Index achieves the fastest drop rate. It always achieves the lowest decoupling error (as shown in the legends in each of the figures). Sometimes Variational Index and Banzhaf show similar performance. We expect that this is because the Banzhaf value is a one-step approximation of Variational Index, and for the specific problem considered, the ranking of the solutions does not change after one-step of fixed point iteration. There are also situations where the rankings of the three criteria are not very distinguishable, however, the specific values are also very different since the decoupling error differs.

## 5.2 EXPERIMENTS ON FEATURE VALUATIONS/ATTRIBUTIONS

We follow the setting of Lundberg and Lee (2017) and reuse the code of `https://github.com/slundberg/shap` with an MIT License. We train classifiers on the Adult dataset[4], which predicts whether an adult's income exceeds 50k dollar per year based on census data. It has 48,842 instances and 14 features such as age, workclass, occupation, sex and capital gain (12 of them used).

**Feature removal results.** This experiment follows a similar fashion as the data removal experiment: we remove the features one by one according to the order defined by the returned criterion, then observe the change of predicted probabilities. Fig. 3 reports the behavior of the three criteria. The first row shows the results from an xgboost classifier (accuracy: 0.893), second row a logistic regression

---

[4]`https://archive.ics.uci.edu/ml/datasets/adult`

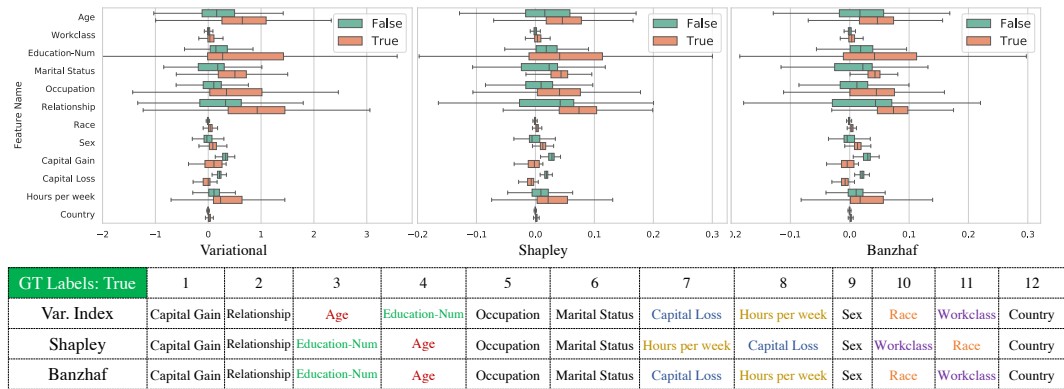

| GT Labels: True | 1 | 2 | 3 | 4 | 5 | 6 | 7 | 8 | 9 | 10 | 11 | 12 |
|---|---|---|---|---|---|---|---|---|---|---|---|---|
| Var. Index | Capital Gain | Relationship | Age | Education-Num | Occupation | Marital Status | Capital Loss | Hours per week | Sex | Race | Workclass | Country |
| Shapley | Capital Gain | Relationship | Education-Num | Age | Occupation | Marital Status | Hours per week | Capital Loss | Sex | Workclass | Race | Country |
| Banzhaf | Capital Gain | Relationship | Education-Num | Age | Occupation | Marital Status | Capital Loss | Hours per week | Sex | Race | Workclass | Country |

Figure 4: Statistics on valuations with the *xgboost* classifier. First row: box plot of valuations. We always consider the predicted probability of the ground truth label. "True" means the samples with positive ground truth label and "False" means with the negative ground truth label. Second row: Average ranking of the 12 features. Colored texts denote different rankings among the three criteria.

classifier (accuracy: 0.842), third row a multi-layer perceptron (accuracy: 0.861). For the probability dropping results, Variational Index usually induces the fastest drop, and it always enjoys the smallest decoupling error, as expected from its mean-field nature. From the waterfall plots, one can see that the three criteria indeed produce different rankings of the features. Take the first row for example. All criteria put "Capital Loss" and "Relationship" as the first two features. However, the remaining features have different ranking: Variational Index and Banzhaf indicate that "Marital Status" should be ranked third, while Shapley ranks it in the fourth position. It is hard to tell which ranking is the best because: 1) There is no golden standard to determine the true ranking of features; 2) Even if there exists a ground truth ranking of some "perfect model", the trained xgboost model here might not be able to reproduce it, since it might not be aligned with the "perfect model".

**Average results.** We further provide the bar plots and averaged ranking across the adult datasets in Fig. 4. From the bar plots one can see that different criterion has slightly different values for each feature on average. Average rankings in the table demonstrate the difference: The three methods do not agree on the colored features, for example, "Age", "Education-Num" and "Captical Loss".

## 5.3 EMPIRICAL CONVERGENCE RESULTS OF ALG. 1

Table 1 shows convergence results of Alg. 1 on feature and data valuation experiments. The value in the cells are the stepwise difference of $\mathbf{x}^k$, $\frac{\|\mathbf{x}^k - \mathbf{x}^{k-1}\|^2}{n}$, which is a classical criterion to measure the convergence of iterative algorithms. One can clearly see that Alg. 1 converges in 5 to 10 iterations.

Table 1: Stepwise difference $\frac{\|\mathbf{x}^k - \mathbf{x}^{k-1}\|^2}{n}$ of Alg. 1 for different experiments.

| Step/Iteration Num | 1 | 2 | 3 | 5 | 9 | 10 |
|---|---|---|---|---|---|---|
| Data Val (breast cancer) | 0.0023 | 3.61e-6 | 1.53e-7 | 2.77e-10 | 9.12e-16 | 0 |
| Data Val (digits) | 0.00099 | 5.93e-7 | 1.46e-8 | 8.92e-12 | 9.25e-18 | 0 |
| Data Val (synthetic) | 0.00059 | 2.49e-8 | 3.13e-10 | 6.06e-14 | 0 | 0 |
| Feature Val (xgboost) | 0.0066 | 1.68e-5 | 8.71e-7 | 2.35e-9 | 1.75e-14 | 9.25e-16 |
| Feature Val (LR) | 0.0092 | 2.63e-5 | 1.44e-6 | 4.31e-9 | 2.14e-15 | 1.28e-16 |
| Feature Val (MLP) | 0.0040 | 4.86e-6 | 1.86e-7 | 2.84e-10 | 6.82e-16 | 3.20e-17 |

DISCUSSIONS AND FUTURE WORK. We have presented an energy-based treatment of cooperative games, in order to improve the valuation problem. It is very worthwhile to explore more in the following directions: 1) Choosing the temperature $T$. The temperature controls the level of fairness since, when $T \to \infty$, all players have equal importance, when $T \to 0$, whereas a player has either 0 or 1 importance (assuming no ties). Perhaps one can use an annealing-style algorithm in order to control the fairness level: starting with a high temperature and gradually decreasing it, one can obtain a series of importance values under different fairness levels. 2) Given the probabilistic treatment of cooperative games, one can naturally add priors over the players, in order to encode more domain knowledge. It may also make sense to consider conditioning and marginalization in light of practical applications. 3) It is very interesting to explore the interaction of a group of players in the energy-based framework, which would result in an "interactive" index among size-$k$ coalitions.

## ETHICS STATEMENT AND BROADER IMPACT

Besides the valuation problems explored in this work, cooperative game theory has already been applied to a wide range of disciplines, to name a few, economics, political science, sociology, biology, so this work could potentially contribute to broader domains as well.

Meanwhile, we have to be aware of possible negative societal impacts, including: 1) negative side effects of the technology itself, for example, possible unemployment issues due to the reduced amount of the need of valuations by human beings; 2) applications in negative downstream tasks, for instance, the data point valuation technique could make it easier to conduct underground transactions of private data.

## REPRODUCIBILITY STATEMENT

All the datasets are publicly available as described in the main text. In order to ensure reproducibility, we have made the efforts in the following respects: 1) Provide code as supplementary material. 2) Provide self-contained proofs of the main claims in Appendices D and E; 3) Provide more details on experimental configurations and experimental results in Appendix G.

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

# Appendix of "Energy-Based Learning for Cooperative Games, with Applications to Valuation Problems in Machine Learning"

CONTENTS

## A  DERIVATIONS OF THE MAXIMUM ENTROPY DISTRIBUTION

One may wonder why do we need energy-based treatment of valuation problems in machine learning? Specifically, under the setting of cooperative games $(N, F(S))$, why we have to take the exponential form of EBM $p(S) \propto \exp(F(S)/T)$? Because one may also formulate it as something else, say, $p(S) \propto (1 + |F(S)|)$?

In one word, because EBM is the maximum entropy distribution. Being a maximum entropy distribution means minimizing the amount of prior information built into the distribution. Another lens to understand it is that: Since the distribution with the maximum entropy is the one that makes the fewest assumptions about the true distribution of data, the principle of maximum entropy can be seen as an application of Occam's razor. Meanwhile, many physical systems tend to move towards maximal entropy configurations over time (Jaynes, 1957a;b).

In the following we will give a derivation to show that $p(S) \propto \exp(F(S)/T)$ is indeed the maximum entropy distribution for a cooperative game. The derivation bellow is closely following Jaynes (1957a;b) for statistical mechanics. Suppose each coalition $S$ is associated with a payoff $F(S)$ with probability $p(S)$. We would like to maximize the entropy $\mathbb{H}(p) = -\sum_{S \subseteq N} p(S) \log p(S)$, subject to the constraints that $\sum_S p(S) = 1, p(S) \geq 0$ and $\sum_S p(S) F(S) = \mu$ (i.e., the average payoff is known as $\mu$).

Writing down the Lagrangian

$$L(p, \lambda_0, \lambda_1) := - \left[ \sum_{S \subseteq N} p(S) \log p(S) + \lambda_0 \left( \sum_{S \subseteq N} p(S) - 1 \right) + \lambda_1 \left( \mu - \sum_{S \subseteq N} p(S) F(S) \right) \right] \quad (16)$$

Setting

$$\frac{\partial L(p, \lambda_0, \lambda_1)}{\partial p(S)} = - \left[ \log p(S) + 1 + \lambda_0 - \lambda_1 F(S) \right] \quad (17)$$

$$= 0 \quad (18)$$

we get:

$$p(S) = \exp[-(\lambda_0 + 1 - \lambda_1 F(S))] \quad (19)$$

$$\lambda_0 + 1 = \log \sum_{S \subseteq N} \exp(\lambda_1 F(S)) =: \log Z \quad (20)$$

is the log-partition function. So,

$$p(S) = \frac{\exp[\lambda_1 F(S))]}{Z} \quad (21)$$

Note that the maximum value of the entropy is

$$H_{\max} = - \sum_{S \subseteq N} p(S) \log p(S) \quad (22)$$

$$= \lambda_0 + 1 - \lambda_1 \sum_{S \subseteq N} p(S) F(S) \quad (23)$$

$$= \lambda_0 + 1 - \lambda_1 \mu \quad (24)$$

So one can get,

$$\lambda_1 = -\frac{\partial H_{\max}}{\partial \mu} =: \frac{1}{T} \quad (25)$$

which defines the inverse temperature.

So we reach the exponential form of $p(S)$ as:

$$p(S) = \frac{\exp[F(S)/T]}{\sum_{S \subseteq N} \exp[F(S)/T]}. \quad (26)$$

## B  COMMON AXIOMS OF VALUATION CRITERIA

Following the definitions in Covert et al. (2020a), the five common axioms are listed as bellow. $\phi_i(F)$ denotes the value assigned to player $i$ in the game $(N, F(S))$. Note that the notions might be slightly different in classical literature of game theory.

**Null player:** For a player $i$ in the cooperative game $(N, F(S))$, if $F(S + i) = F(S)$ holds for all $S \subseteq N - i$, then its value should be $\phi_i(F) = 0$.

**Symmetry:** For any two players $i, j$ in the cooperative game $(N, F(S))$, if $F(S + i) = F(S + j)$ holds for all $S \subseteq N - i - j$, then it holds that $\phi_i(F) = \phi_j(F)$.

**Marginalism:** For two games $(N, F(S))$ and $(N, G(S))$ where all players have identical marginal contributions, the players obtain equal valuations: $F(S + i) - F(S) = G(S + i) - G(S)$ holds for all $(i, S)$, then it holds that $\phi_i(F) = \phi_i(G)$.

**Additivity:** For two games $(N, F(S))$ and $(N, G(S))$, if they are combined, the total contribution of a player is equal to the sum of its individual contributions on each game: $\phi_i(F+G) = \phi_i(F) + \phi_i(G)$.

**Efficiency:** The values add up to the difference in value between the grand coalition and the empty coalition: $\sum_{i \in N} \phi_i(F) = F(N) - F(\emptyset)$.

**Remark 1** (The notion of "marginalism"). *Specifically, the marginalism axiom was first mentioned by (Young, 1985, Equation 7 on page 70), where it was called the "independence" condition; Following (Chun, 1989, page 121), it was formally called the "marginality". This axiom requires a player's payoffs to depend only on his own marginal contributions – whenever they remain unchanged, his payoffs should be unaffected.*

---

**Algorithm 2:** NAIVE MEAN FIELD FOR CALCULATING THE VARIATIONAL INDEX

---

**Input:** A cooperative game $(N, F(S))$ with $n$ players. Initial marginal $\mathbf{x}^0 \in [0, 1]^n$; #epochs $K$.
**Output:** The Variational Index $\mathbf{s}^* = \sigma^{-1}(\mathbf{x}^*)$

1   $\mathbf{x} \leftarrow \mathbf{x}^0$;
2   **for** *epoch from* $1$ *to* $K$ **do**
3     **for** $i = 1 \rightarrow n$ **do**
4       let $v_i$ be the player being operated;
5       $x_{v_i} \leftarrow \sigma(\nabla_{v_i} f_{\text{mt}}^F(\mathbf{x})/T) = (1 + \exp(-\nabla_{v_i} f_{\text{mt}}^F(\mathbf{x})/T)^{-1}$ ;

6   $\mathbf{x}^* \leftarrow \mathbf{x}$;

---

## C  THE NAIVE MEAN FIELD ALGORITHM

The naive mean field algorithm is one of the most classical algorithm for mean field inference. It is summarized in Alg. 2.

## D  PROOF OF RECOVERING CLASSICAL CRITERIA

For Banzhaf value, by comparing its definition in Eq. (2) with Eq. (13) it reads,

$$\text{Ba}_i = \sum_{S \subseteq N-i} [F(S+i) - F(S)] \frac{1}{2^{n-1}} = \nabla_i f_{\text{mt}}^F(0.5 * \mathbf{1}) = T\sigma^{-1}(\text{MFI}(0.5 * \mathbf{1}; 1)), \quad (27)$$

which is the 1-step variational value initialied at $0.5 * \mathbf{1}$.

For Shapley value, according to Grabisch et al. (2000), here we prove a stronger conclusion regarding the generalization of Shapley value: Shapley interaction index, which is defined for any coalition $S$:

$$\text{Sh}_S = \sum_{T \subseteq N-S} \frac{(n - |T| - |S|)!|T|!}{(n - |S| + 1)!} \sum_{L \subseteq S} (-1)^{|S|-|L|} F(L + T). \quad (28)$$

Given Hammer and Rudeanu (2012), we have a second form of the multilinear extension as:

$$f_{\text{mt}}^F(\mathbf{x}) = \sum_{S \subseteq N} a(S) \prod_{i \in S} x_i, \mathbf{x} \in [0, 1]^n, \quad (29)$$

where $a(S) := \sum_{T \subseteq S} (-1)^{|S|-|T|} F(T)$ is the Mobius transform of $F(S)$.

Then, one can show the $S$-derivative of $f_{\text{mt}}^F(\mathbf{x})$ is (suppose $S = \{i_1, ..., i_{|S|}\}$),

$$\Delta_S f_{\text{mt}}^F(\mathbf{x}) := \frac{\partial^{|S|} f_{\text{mt}}^F(\mathbf{x})}{\partial x_{i_1}, ..., \partial x_{i_{|S|}}} = \sum_{T \supseteq S} a(T) \prod_{i \in T-S} x_i. \quad (30)$$

So,

$$\Delta_S f_{\text{mt}}^F(x\mathbf{1}) = \sum_{T \supseteq S} a(T) x^{|T|-|S|}. \quad (31)$$

Then it holds,

$$\int_0^1 \Delta_S f_{\text{mt}}^F(x\mathbf{1}) dx = \int_0^1 \sum_{T \supseteq S} a(T) x^{|T|-|S|} dx \quad (32)$$

$$= \sum_{T \supseteq S} a(T) \int_0^1 x^{|T|-|S|} dx \tag{33}$$

$$= \sum_{T \supseteq S} a(T)(|T| - |S| + 1)^{-1} \tag{34}$$

According to Grabisch (1997), we have $\mathrm{Sh}_S = \sum_{T \supseteq S} a(T)(|T| - |S| + 1)^{-1}$, then we reach the conclusion:

$$\mathrm{Sh}_S = \int_0^1 \Delta_S f_{\mathrm{mt}}^F(x\mathbf{1}) dx. \tag{35}$$

When $|S| = 1$, we recover the conclusion for Shapley value.

## E  PROOF OF THEOREM 1

**Theorem 1** (Axiomatisation of $K$-Step Variational Values of Def. 3). *If initialized uniformly, i.e.,* $\mathbf{x}^0 = x\mathbf{1}, x \in [0,1]$, *all the variational values in the trajectory* $T\sigma^{-1}(MFI(\mathbf{x}; k)), k = 1, 2, 3...$ *satisfy the null player, marginalism and symmetry axioms.*

*Proof of Theorem 1.* In step $k$, we know that the value to player $i$ is:

$$T\sigma^{-1}(\mathrm{MFI}(\mathbf{x}; k))_i = \sum_{S \subseteq N-i} [F(S+i) - F(S)] \prod_{j \in S} x_j \prod_{j' \in N-S-i} (1 - x_{j'}) \tag{36}$$

For the **null player** property, since $F(S + i) = F(S)$ always holds, it is easy to see that $T\sigma^{-1}(\mathrm{MFI}(\mathbf{x}; k))_i = 0$ holds for all $i \in N$.

Now we will show that the **symmetry** property holds. The value to player $i'$ is:

$$T\sigma^{-1}(\mathrm{MFI}(\mathbf{x}; k))_{i'} = \sum_{S' \subseteq N-i'} [F(S'+i') - F(S')] \prod_{j \in S'} x_j \prod_{j' \in N-S'-i} (1 - x_{j'}) \tag{37}$$

Now let us compare different terms in the summands of Eq. (36) and Eq. (37). We try to match the summands one by one. There are two situations:

**Situation I:** For any $S \subseteq N - i - i'$, we choose $S' = S$.

In this case we have $F(S + i) - F(S) = F(S' + i') - F(S')$. For the products of $\mathbf{x}$ we have:

$$\prod_{j \in S} x_j \prod_{j' \in N-S-i} (1 - x_{j'}) - \prod_{j \in S'} x_j \prod_{j' \in N-S'-i} (1 - x_{j'}) = \tag{38}$$

$$\prod_{j \in S} x_j \prod_{j' \in N-S-i-i'} (1 - x_{j'})[(1 - x_{i'}) - (1 - x_i)] \tag{39}$$

We know that $x_{i'} = x_i$ holds from step 0, by simple induction, we know that $x_{i'} = x_i$ holds for step $k$ as well. So in this situation, the summands equal to each other.

**Situation II:** For any $S = A + i'$, we choose $S' = A + i$, where $A \subseteq N - i - i'$. In this case, it still holds that $F(S + i) - F(S) = F(S' + i') - F(S')$. For the products of $\mathbf{x}$ we have:

$$\prod_{j \in S} x_j \prod_{j' \in N-S-i} (1 - x_{j'}) - \prod_{j \in S'} x_j \prod_{j' \in N-S'-i'} (1 - x_{j'}) = \tag{40}$$

$$\prod_{j \in A} x_j \prod_{j' \in N-A-i-i'} (1 - x_{j'})[x_{i'} - x_i]. \tag{41}$$

Again, by the simple induction, we know that $x_{i'} = x_i$ holds for step $k$.

The above two situations finishes the proof of symmetry.

For the **marginalisim** axiom, one can see that the update step for the two games are identical, and it is easy to deduce that they produce exactly the same trajectories, given that they have the same initializations.

$\square$

# F   MISCELLANEOUS RESULTS IN SEC. 4

**Gradient of entropy in Sec. 4.2**   Note that $q$ is a fully factorized product distribution $q(S; \mathbf{x}) := \prod_{i \in S} x_i \prod_{j \notin S}(1 - x_j), \mathbf{x} \in [0, 1]^n$, so its entropy $\mathbb{H}(q(S; \mathbf{x}))$ can be written as the sum of entropy of $n$ independent Bernoulli distributions. And the entropy of one Bernoulli distribution with parameter $x_i$ is

$$-x_i \log x_i - (1 - x_i) \log(1 - x_i)$$

So we have,

$$\nabla_i \mathbb{H}(q(S; \mathbf{x})) = \nabla_i \sum_{i=1}^{n} [-x_i \log x_i - (1 - x_i) \log(1 - x_i)] \tag{42}$$

$$= \nabla_i [-x_i \log x_i - (1 - x_i) \log(1 - x_i)] \tag{43}$$

$$= \log \frac{1 - x_i}{x_i} \tag{44}$$

**Satisfying more axioms is not essential for valuation problems.**   Notably, in cooperative game theory, one line of work is to seek for solution concepts that would satisfy more axioms. However, for valuation problems in machine learning, this is arguably not essential. For example, similar as what Ridaoui et al. (2018) argues, efficiency does not make sense for certain games.

For a simple illustration, let us consider a voting game from a classification model with 3 binary features $\mathbf{x} \in \{0, 1\}^3$ with weights $\mathbf{w} = [2, 1, 1]^\top$: $f(\mathbf{x}) := \mathbb{1}_{\{\mathbf{w}^\top \mathbf{x} \geq 3\}}$. Now we are trying to find the valuation of each feature in $N = \{x_1, x_2, x_3\}$. Naturally, the value function in the corresponding voting game shall be $F(S) = f(\mathbf{x}_S)$ where $\mathbf{x}_S$ means setting the coordinates of $\mathbf{x}$ inside $S$ to be 1 while leaving others to be 0. In this game let us count how many times each feature could flip the classification result: for feature $x_1$, there are three situations: $F(\{1, 2\}) - F(\{2\})$, $F(\{1, 3\}) - F(\{3\})$ and $F(\{1, 2, 3\}) - F(\{2, 3\})$; for feature $x_2$, there are one situation: $F(\{1, 2\}) - F(\{1\})$; for feature $x_3$, there are one situation: $F(\{1, 3\}) - F(\{1\})$. Then the voting power (or valuation) of each feature shall follows a $3 : 1 : 1$ ratio. By simple calculations, one can see that the Banzhaf values of the three features are $\frac{3}{4}, \frac{1}{4}, \frac{1}{4}$, which is consistent with the ratio of the expected voting power. However, the Shapley values of them are $\frac{4}{6}, \frac{1}{6}, \frac{1}{6}$, which is not consistent due to satisfying the efficiency axiom.

By the above example we are trying to explain that for valuation problems, satisfying more axioms is not necessary, sometimes even does not make sense. Whether more axioms shall be considered and which sets of them shall be added really depend on the specific scenario, which will be left for important future work.

**The "one-shot sampling trick" to accelerate Alg. 1.**   Indeed, Variational Index needs 5 to 10 iterations to converge. In each iteration one has to evaluate the gradient of multilinear extension $\nabla f_{\text{mt}}^F(\mathbf{x})$, which needs MCMC sampling to estimate the exponential sum.

Here we suggest a "one-shot sampling trick", when it is expensive to evaluate the value function $F(S)$. This trick could reuse the sampled values in each iteration, such that Alg. 1 could run with the similar cost as calculating Banzhaf values.

The one-shot sampling trick is built upon one formulation taken from Eq. (13):

$$\nabla_i f_{\text{mt}}^F(\mathbf{x}) = \sum_{S \subseteq N-i} [F(S+i) - F(S)] q(S; (\mathbf{x}|x_i \leftarrow 0))$$

For coordinate $i$ of $\nabla f_{\text{mt}}^F(\mathbf{x})$, we can firstly sample $m$ coalitions uniformly randomly from $2^{N-i}$, and evaluate their marginal contributions. Then in each of the following iterations, one could reuse the one-shot sampled marginal contributions to estimate the gradient according to the above equation. In this way, we could make the cost of multi-step running of Alg. 1 similar as that of Banzhaf value in

terms of the number of $F(S)$ evaluations. Compared to the original iterative sampling from $q$, this one-shot sampling might come with a variance-complexity tradeoff, for which we will explore as a future work.

## G MORE CONFIGURATION DETAILS AND EXPERIMENTAL RESULTS

### G.1 SYNTHETIC EXPERIMENTS ON SUBMODULAR (FLID) GAMES

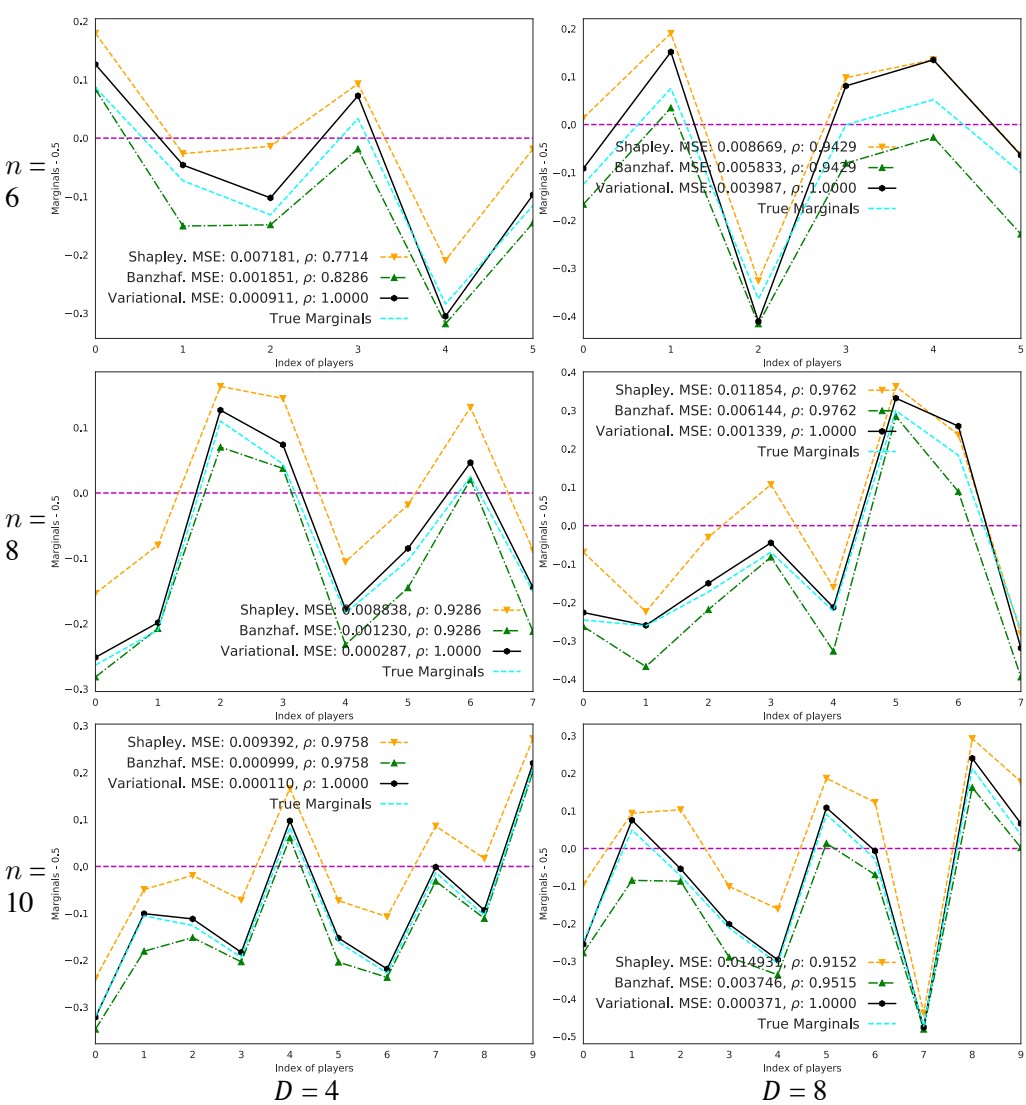

Figure 5: Comparison of different importance measures for FLID games. Hidden dimensions: First column $D$=4, second column $D$=8. $n$ denotes # of players. Vertical lines means (marginals - 0.5) since we would like to clearly show positive players (marginal > 0.5) and negative players (marginal < 0.5).

Here we define a synthetic game with the value function as a FLID (Tschiatschek et al., 2016) objective $F(S)$, which is a diversity boosting model satisfying the submodularity property. We know that its multilinear extension admits polynomial time algorithms (Bian et al., 2019). Let $\mathbf{W} \in \mathbb{R}_+^{|N| \times D}$ be the weights, each row corresponds to the latent representation of an item, with $D$ as

the dimensionality. Then

$$F(S) := \sum_{i \in S} u_i + \sum_{d=1}^{D} \left( \max_{i \in S} W_{i,d} - \sum_{i \in S} W_{i,d} \right) = \sum_{i \in S} u'_i + \sum_{d=1}^{D} \max_{i \in S} W_{i,d}, \quad (45)$$

which models both coverage and diversity, and $u'_i = u_i - \sum_{d=1}^{D} W_{i,d}$. In order to test the performance of the proposed variational objectives, we consider small synthetic games with 6, 8 and 10 players such that the ground truth marginals can be computed exhaustively. We would like to compare with the true marginals $p(i \in \mathbf{S})$ since they represent the probability that player $i$ participates in all coalitions, which is hard to compute in general. The distance to the true marginals is also a natural measure of the decoupling error as defined in Def. 1. We apply a sigmoid function to Shapley value and Banzhaf value in order to translate them to probabilities. We calculate the mean squared error (MSE) and Spearman's rank correlation ($\rho$) to the ground truth marginals $p(i \in \mathbf{S})$ and report them in the figure legend. Fig. 5 collects the figures, one can see that the Variational Index clearly obtains better performance in terms of MSE and Spearman's rank correlation compared to the one-point solutions (Shapley value and Banzhaf value) in all experiments.

## G.2 Details of the Models for Feature Valuations

For xgboost, the train accuracy is 0.8934307914376094, the specific configuration with the xgboost package is:

```
XGBClassifier(base_score=0.5, booster='gbtree', colsample_bylevel=1,
              colsample_bynode=1, colsample_bytree=1, gamma=0, gpu_id=-1,
              importance_type='gain', interaction_constraints='',
              learning_rate=0.300000012, max_delta_step=0, max_depth=6,
              min_child_weight=1, missing=nan, monotone_constraints='()',
              n_estimators=100, n_jobs=56, num_parallel_tree=1,
              random_state=0, reg_alpha=0, reg_lambda=1,
              scale_pos_weight=1, subsample=1, tree_method='exact',
              validate_parameters=1, verbosity=None)
```

We used the sklearn package for the logistic regression and MLP classifiers. For logistic regression, the accuracy is 0.8418967476428857, the specific configuration is:

```
LogisticRegression(random_state=0, solver="liblinear", C=0.5)
```

For the MLP, its accuracy is 0.8614600288688923, and the configuration is:

```
MLPClassifier(random_state=0, max_iter=300,
              learning_rate_init=0.002,
              hidden_layer_sizes=(50,50))
```

## G.3 More Results on Feature Valuations

In this part we provide more results on feature removal in Figures 6 and 7. Fig. 6 shows similar behavior as that shown in the main text. Fig. 7 provides not very distinguishable results of the three criteria.

## G.4 More Average Results on Feature Valuations

We provide additional statistical results with the MLP model (Fig. 9) and logistic regression model (Fig. 10). Meanwhile, we also put the full statistics on the xgboost model in Fig. 8 since in the main text the table of the samples with "False" grouthtruth label is skipped due to space limit. From Fig. 9 one can observe that Variational Index induces different rankings of the features compared to Shapley and Banzhaf: Variational Index ranks "Marital Status" the second, while Shapley and Banzhaf put it in the third location.

It is also very interesting to see that the logistic regression model (with the lowest training accuracy among the three models, shown in Fig. 10) provides different ranking for the first two features compared to MLP and xgboost. For the samples with "True" groundtruth labels, "Education-Num" is the first important feature for the logistic regression model, while "Captical Gain" was ranked first for the MLP and xgboost.

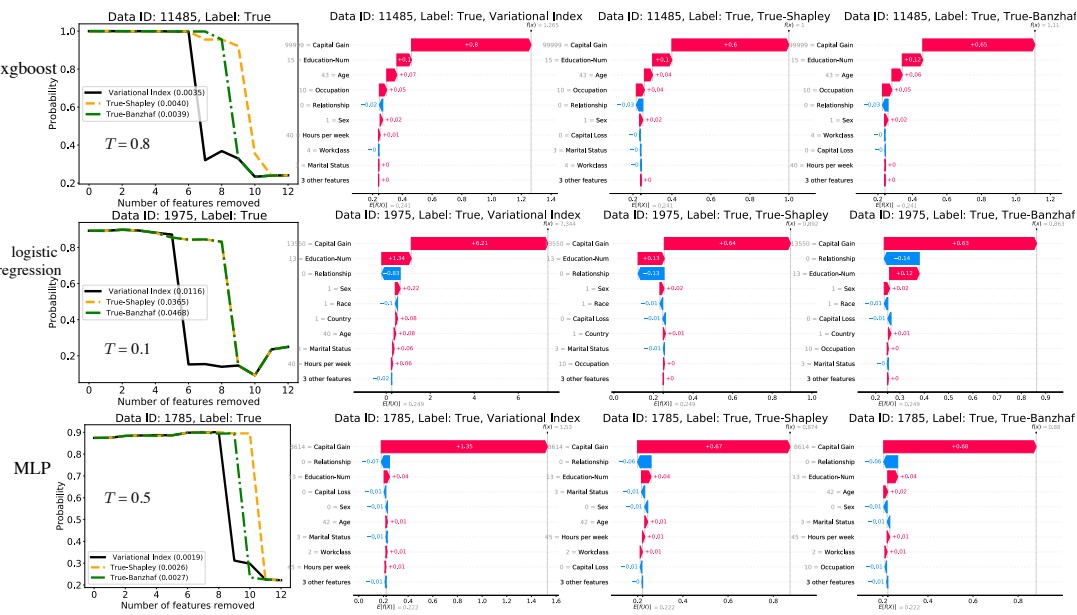

Figure 6: First column: Change of predicted probabilities when removing features. The *decoupling error* is included in the legend. Last three columns: waterfall plots of feature importance from Variational Index, Shapley and Banzhaf.

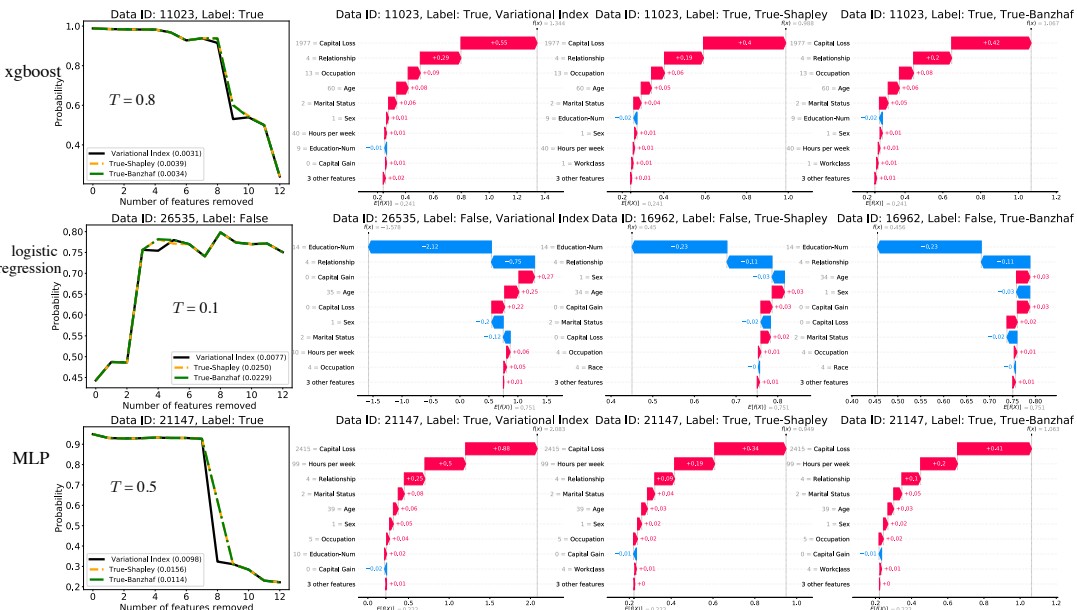

Figure 7: Not very distinguishable results. First column: Change of predicted probabilities when removing features. The *decoupling error* is included in the legend. Last three columns: waterfall plots of feature importance from Variational Index, Shapley and Banzhaf.

## G.5 EXPERIMENTS WITH MORE PLAYERS

Furthermore, we experiment with a bit larger number of players ($n = 80$) using MCMC sampling to approximate the partial derivative in Eq. (13). The sampling based approximation works pretty fast. Table 2 shows the top 15 ranked players returned by our method, Shapley value and Banzhaf value for a synthetic data valuation problem. Note that the ranking of Variational Index is more similar to that of Banzhaf value than that of Shapley value.

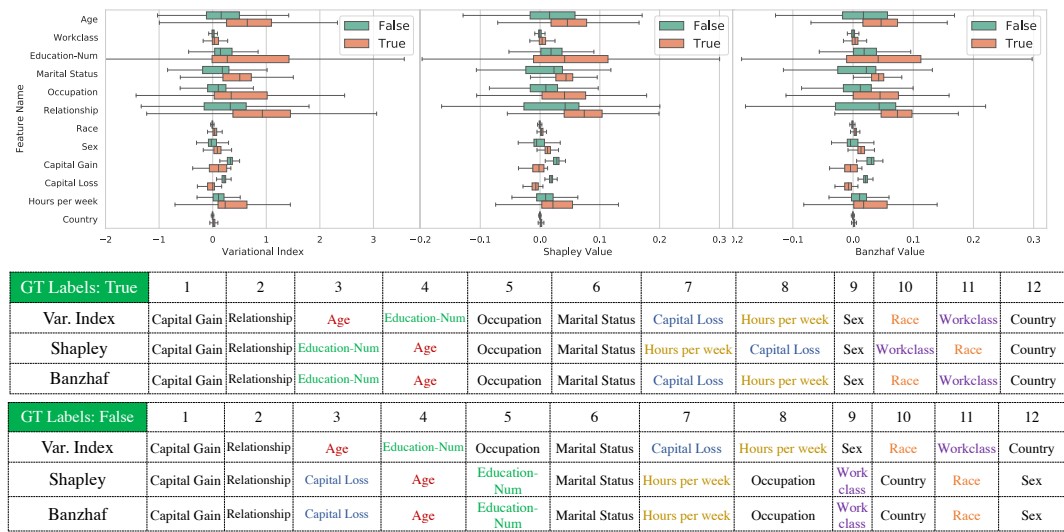

| GT Labels: True | 1 | 2 | 3 | 4 | 5 | 6 | 7 | 8 | 9 | 10 | 11 | 12 |
|---|---|---|---|---|---|---|---|---|---|---|---|---|
| Var. Index | Capital Gain | Relationship | Age | Education-Num | Occupation | Marital Status | Capital Loss | Hours per week | Sex | Race | Workclass | Country |
| Shapley | Capital Gain | Relationship | Education-Num | Age | Occupation | Marital Status | Hours per week | Capital Loss | Sex | Workclass | Race | Country |
| Banzhaf | Capital Gain | Relationship | Education-Num | Age | Occupation | Marital Status | Capital Loss | Hours per week | Sex | Race | Workclass | Country |

| GT Labels: False | 1 | 2 | 3 | 4 | 5 | 6 | 7 | 8 | 9 | 10 | 11 | 12 |
|---|---|---|---|---|---|---|---|---|---|---|---|---|
| Var. Index | Capital Gain | Relationship | Age | Education-Num | Occupation | Marital Status | Capital Loss | Hours per week | Sex | Race | Workclass | Country |
| Shapley | Capital Gain | Relationship | Capital Loss | Age | Education-Num | Marital Status | Hours per week | Occupation | Workclass | Country | Race | Sex |
| Banzhaf | Capital Gain | Relationship | Capital Loss | Age | Education-Num | Marital Status | Hours per week | Occupation | Workclass | Country | Race | Sex |

Figure 8: Full statistics on valuations with the *xgboost* classifier (in the main text the last row is skipped due to space limit). First row: box plot of valuations returned by the three criteria. We always consider the predicted probability of the ground truth label. "True" means the samples with positive ground truth label and "False" means with the negative ground truth label. Second and third rows: Average ranking of the 12 features. Colored texts denote different rankings among the three criteria.

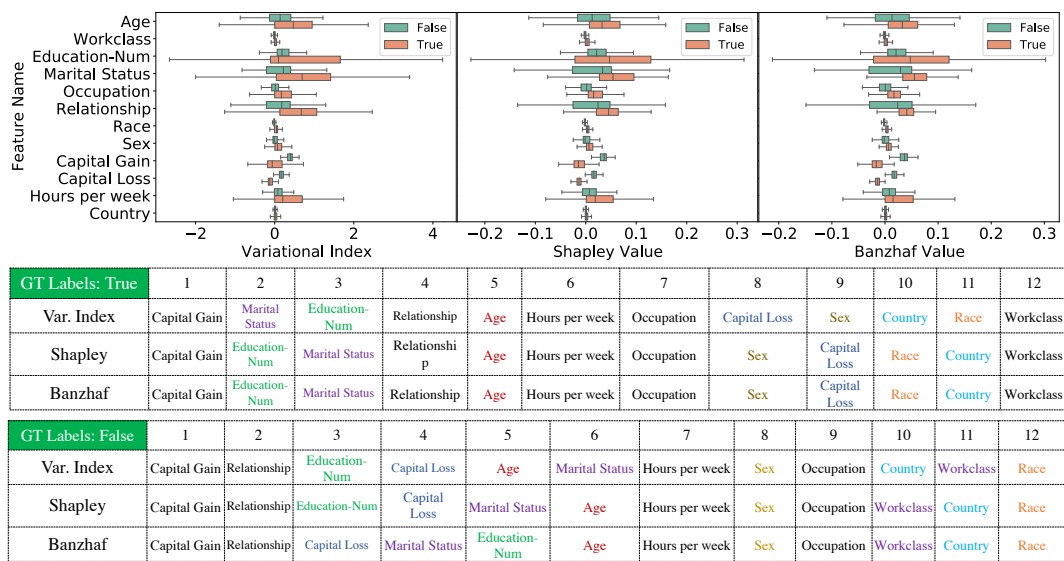

| GT Labels: True | 1 | 2 | 3 | 4 | 5 | 6 | 7 | 8 | 9 | 10 | 11 | 12 |
|---|---|---|---|---|---|---|---|---|---|---|---|---|
| Var. Index | Capital Gain | Marital Status | Education-Num | Relationship | Age | Hours per week | Occupation | Capital Loss | Sex | Country | Race | Workclass |
| Shapley | Capital Gain | Education-Num | Marital Status | Relationship | Age | Hours per week | Occupation | Sex | Capital Loss | Race | Country | Workclass |
| Banzhaf | Capital Gain | Education-Num | Marital Status | Relationship | Age | Hours per week | Occupation | Sex | Capital Loss | Race | Country | Workclass |

| GT Labels: False | 1 | 2 | 3 | 4 | 5 | 6 | 7 | 8 | 9 | 10 | 11 | 12 |
|---|---|---|---|---|---|---|---|---|---|---|---|---|
| Var. Index | Capital Gain | Relationship | Education-Num | Capital Loss | Age | Marital Status | Hours per week | Sex | Occupation | Country | Workclass | Race |
| Shapley | Capital Gain | Relationship | Education-Num | Capital Loss | Marital Status | Age | Hours per week | Sex | Occupation | Workclass | Country | Race |
| Banzhaf | Capital Gain | Relationship | Capital Loss | Marital Status | Education-Num | Age | Hours per week | Sex | Occupation | Workclass | Country | Race |

Figure 9: Statistics on valuations with the *MLP* classifier. First row: box plot of valuations returned by the three criteria. We always consider the predicted probability of the ground truth label. "True" means the samples with positive ground truth label and "False" means with the negative ground truth label. Second and third rows: Average ranking of the 12 features. Colored texts denote different rankings among the three criteria.

## G.6 EFFECT OF NUMBER OF SAMPLES IN MCMC SAMPLING

Here we illustrate the accuracy sampling tradeoff when estimating the gradient of multilinear extension using MCMC sampling. The results is shown in Fig. 11. One can observe that with more samples, Alg. 1 will converge faster (in fewer number of epochs).

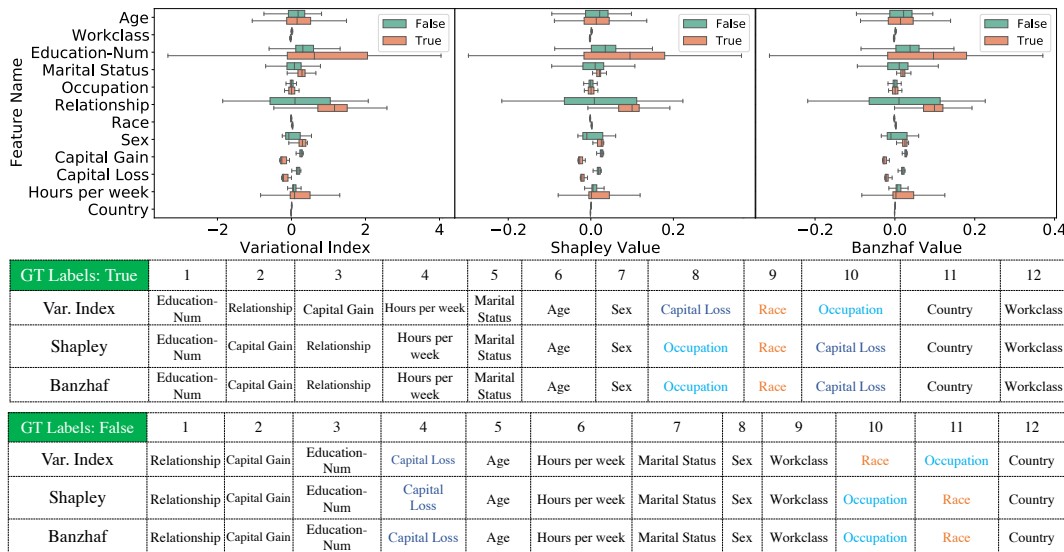

| GT Labels: True | 1 | 2 | 3 | 4 | 5 | 6 | 7 | 8 | 9 | 10 | 11 | 12 |
|---|---|---|---|---|---|---|---|---|---|---|---|---|
| Var. Index | Education-Num | Relationship | Capital Gain | Hours per week | Marital Status | Age | Sex | Capital Loss | Race | Occupation | Country | Workclass |
| Shapley | Education-Num | Capital Gain | Relationship | Hours per week | Marital Status | Age | Sex | Occupation | Race | Capital Loss | Country | Workclass |
| Banzhaf | Education-Num | Capital Gain | Relationship | Hours per week | Marital Status | Age | Sex | Occupation | Race | Capital Loss | Country | Workclass |

| GT Labels: False | 1 | 2 | 3 | 4 | 5 | 6 | 7 | 8 | 9 | 10 | 11 | 12 |
|---|---|---|---|---|---|---|---|---|---|---|---|---|
| Var. Index | Relationship | Capital Gain | Education-Num | Capital Loss | Age | Hours per week | Marital Status | Sex | Workclass | Race | Occupation | Country |
| Shapley | Relationship | Capital Gain | Education-Num | Capital Loss | Age | Hours per week | Marital Status | Sex | Workclass | Occupation | Race | Country |
| Banzhaf | Relationship | Capital Gain | Education-Num | Capital Loss | Age | Hours per week | Marital Status | Sex | Workclass | Occupation | Race | Country |

Figure 10: Statistics on valuations with the *logistic regression* classifier. First row: box plot of valuations returned by the three criteria. We always consider the predicted probability of the ground truth label. "True" means the samples with positive ground truth label and "False" means with the negative ground truth label. Second and Third rows: Average ranking of the 12 features. Colored texts denote different rankings among the three criteria.

Table 2: Indices of the top 15 ranked players returned by different methods.

| Rank of players | 1 | 2 | 3 | 4 | 5 | 6 | 7 | 8 | 9 | 10 | 11 | 12 | 13 | 14 | 15 |
|---|---|---|---|---|---|---|---|---|---|---|---|---|---|---|---|
| Variational Index | 9 | 13 | 11 | 3 | 54 | 7 | 18 | 36 | 32 | 42 | 46 | 40 | 6 | 18 | 23 |
| Banzhaf value | 9 | 13 | 11 | 3 | 54 | 18 | 7 | 32 | 46 | 36 | 2 | 27 | 40 | 17 | 10 |
| Shaplay Value | 52 | 33 | 27 | 1 | 4 | 58 | 32 | 14 | 42 | 46 | 40 | 6 | 18 | 23 | 47 |

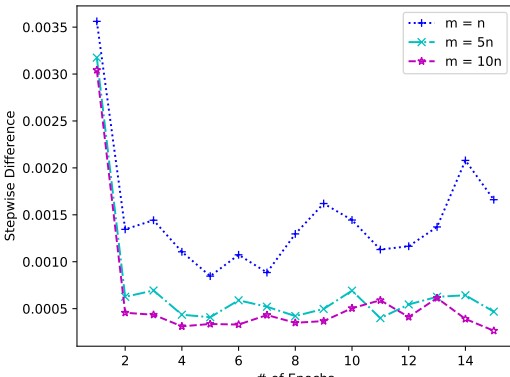

Figure 11: Stepwise difference with difference number of samples $m$ when estimating the gradient of multilinear extension using MCMC sampling. The three curves: $m = n$, $m = 5n$, $m = 10n$.

