# OpenReview forum: "Energy-Based Learning for Cooperative Games, with Applications to Valuation Problems in Machine Learning"
_ICLR.cc/2022/Conference — ICLR 2022 Poster_

### Official Review · Reviewer_5RYD · 2021-10-24

**Correctness:** 4
**Technical Novelty And Significance:** 2
**Empirical Novelty And Significance:** 2
**Recommendation:** 6
**Confidence:** 4

**Main Review:**

Strengths:
- The authors show that one can derive the Shapley value and Banzhaf index using one step of gradient ascent using specific initializations. I think it’s nice that these two criteria can be unified under this model.
- The paper seemed polished and was easy to read.

Weaknesses:
- I think that the main way this paper could be improved is that I couldn’t quite understand the benefit of this approach to defining the importance scores over classic approaches like the Shapley value and the Banzhaf index. It’s not any easier to compute and at least according to what’s written in the paper thus far, it doesn’t seem to satisfy any additional game-theoretic axioms than the Shapley value or Banzhaf index. Lastly, in the experiments, it seems to perform about as well as these other two indices; the experiments don’t seem illustrate its superiority as decisively as one might hope.
- There are a few ways that the model choices could be further justified. The paper is really built around the specific choice $p$ of the distribution over coalitions, which is justified in a few sentences in the third paragraph of the introduction. However, since the reader really has to buy in to this choice of $p$ to appreciate the rest of the paper, I think that a bit more justification would be useful. One other thing I couldn’t quite follow is why the importance score ends up being defined in terms of the inverse sigma functions.

Smaller comments:
- 3rd paragraph of Section 1: How is mu chosen? Why doesn’t it show up in Equation (1)?
- 4th paragraph of Section 1: At first, I was confused about what the “valuation problem” is. Then I realized it’s the problem of defining the importance vector; it would be helpful to clarify this.
- Last paragraph of Section 1: I have a few suggestions:
   - I didn’t know what was meant by a “decoupling perspective” until I got to Section 4, so I wasn’t able to understand this sentence.
   - “Intriguing properties” is a bit too vague, in my opinion. After reading Section 5, I’m not really sure what these intriguing properties would be, so it would definitely help to spell these out.


**Summary Of The Paper:**

This paper studies valuation problems from cooperative game theory. There are $n$ agents and a valuation function $F: [n] \to R$ where $F(S)$ is the collective payoff of the coalition $S \subseteq [n]$. The goal is to use this function $F$ to define an importance vector $\phi(F) \in R^n$. Examples include the Shapley value and Banzhaf index.

The authors introduce a probabilistic treatment of this problem, where they use $F$ to define a probability distribution $p$ where $p(S)$ is the probability that coalition $S$ forms. They then phrase the problem of defining an importance vector $\phi(F)$ as a decoupling problem. Under $p$, the $n$ agents may be correlated in a complicated way, but to assign each of them an individual importance value, one must decouple their interactions, or simplify their correlations. The goal is then to find a product distribution $q$ that is as close to $p$ as possible under the KL divergence. Specifically, the authors define $q$ to be an $n$ independent Bernoulli distribution, where the probability that agent $i$ participates in the coalition is denoted $x_i$. The authors show how to optimize the probabilities $x_1, \dots, x_n$ using coordinate ascent. Finally, they define the importance score of player $i$ as $\log(x_i/(1-x_i))$ (ignoring a temperature $T$ term for simplicity). The authors show that the resulting importance vector satisfies many of the game-theoretic axioms that the Shapley value and Banzhaf index satisfy, like the null player, marginalism, and symmetric axioms.

In the experiments, the authors look at small instances with $n = 25$ where it is actually possible to compute the gradients exactly (as opposed to an approximate sampling method). The applications they look at are for data valuation and feature attribution in the context of machine learning. For these tasks, they show that their proposed approach performs about the same as the Shapley value and Banzhaf index, and sometimes a bit better.

**Summary Of The Review:**

Overall, I think that this paper was nicely written and I like that it unifies the Shapley value and Banzhaf index, but I don’t yet see why the proposed approach is necessarily better than using either of these two existing criteria, so I’m leaning towards rejection.

---

> ### Author Response · Authors · 2021-11-16
> **Answers to Reviewer (Part 1)**
>
>
> We thank the reviewer for the extensive comments that help us to improve our work. Note that  there may be several misunderstandings regarding this work, for
> which we will clarify in detail below.
>
>
> ### Comments: Benefits of the proposed maximum entropy framework, axioms & empirical results
>
> > I think that the main way this paper could be improved is that I
> couldn’t quite understand the benefit of this approach to defining the importance scores over classic approaches like the Shapley value and the Banzhaf index. It’s not any easier to compute and at least
> according to what’s written in the paper thus far, it doesn’t seem
> to satisfy any additional game-theoretic axioms than the Shapley value or Banzhaf index. Lastly, in the experiments, it seems to perform about
> as well as these other two indices; the experiments don’t seem illustrate
>  its superiority as decisively as one might hope.
>
> **ANSWER**:
>
> There may be several misunderstandings regarding the advantages of this work. So before re-summarizing the benefits of our maximum entropy framework, we would like to clarify the following regarding the benefits of our methods:
>
> **1. Satisfying more axioms is not essential for valuation problems.**
>
> By the bellow example we  will explain that for valuation problems, satisfying more axioms is not necessary, sometimes even does not make sense. Whether more axioms shall be considered and which sets of them shall be added really depend on the specific scenario being modeled. Similar arguments were also discussed in recent papers, such as [Karczmarz, Adam, Anish Mukherjee, Piotr Sankowski, and Piotr Wygocki. "Improved Feature Importance Computations for Tree Models: Shapley vs. Banzhaf." arXiv preprint arXiv:2108.04126 (2021).].
>
> According to Theorem 1, in general,   our proposed $K$-step variational values satisfy the minimal  set of axioms in order to being a suitable valuation criterion. Note that specific realizations of the $K$-step variational values could also satisfy more axioms, for example,
> the $1$-step variational value initialied at $0.5*\mathbf{1}$
>  also satisfies the additivity axiom. Furthermore, we have the following observations:
>
>  Notably, in cooperative game theory, one line of work is to seek for solution concepts that would satisfy more axioms. However, for valuation problems in machine learning, this is arguably not essential.  For example, similar as what [Ridaoui et al 2018] argues,  efficiency   does not make sense for certain games. For a simple illustration,  let us consider a  voting game from a  classification model with 3 binary features $\mathbf{x}\in \\{0, 1\\}^3$ with weights $\mathbf{w} = [2, 1, 1]^T$:
> $f(\mathbf{x}) := \text{Indicator}_{\{\mathbf{w}^T \mathbf{x} \geq 3 \}}$.  Now we are trying to find the valuation of each feature in $V = \\{x_1, x_2, x_3\\}$.
> Naturally, the value function in the corresponding voting game shall be $F(S) = f(\mathbf{x}_S)$ where $\mathbf{x}_S$ means setting the coordinates of $\mathbf{x}$ inside $S$ to be 1 while leaving others to be 0.
> In this game let us count  how many times each feature could flip the classification result: for feature $x_1$, there are three situations: $F(\{1,2\}) - F(\{2\})$,   $F(\{1,3\}) - F(\{3 \})$ and
> $F(\{1,2, 3\}) - F(\{2, 3 \})$; for feature $x_2$, there are one situation: $F(\{1,2\}) - F(\{1\})$; for feature $x_3$, there are one situation: $F(\{1,3\}) - F(\{1\})$. Then the voting power (or valuation) of each feature shall follow a $3:1:1$ ratio.
> By simple calculations, one can see that the Banzhaf indices of the three features are $\frac{3}{4}, \frac{1}{4}, \frac{1}{4}$, which is consistent with the ratio of the expected voting power. However, the Shapley values of them are $\frac{4}{6}, \frac{1}{6}, \frac{1}{6}$, which is not consistent due  to satisfying the efficiency axiom.
>
>
> **2. Scalability is a common issue for all game-theoretic valuation
> criteria.**
>
> As discussed in the last paragraph of Sec. 4.2, all of them
> need to evaluate the gradient of multilinear extension. MCMC sampling
> could help with approximating them.   We have also empirically verified
> that Alg. 1 converges fast within 5 ~ 10 steps in Sec. 5.3.   Meanwhile, we have also come up
> with a "one-shot sampling trick" which could potentially make Alg. 1 run with the similar cost as calculating Banzhaf values, when
> evaluating the value function $F(S)$ is expensive.

---

> > ### Author Response · Authors · 2021-11-16
> > **Answers to Reviewer (Part 2)**
> >
> > (continued for the previous comment)
> >
> >
> > **3. Superior Empirical results in terms of lower decoupling error and better valuation performance.**
> >
> > 1. Decoupling error (see Def. 1) is a clear performance metric to measure the decoupling performance.  The proposed variational index always enjoys lower decoupling error compared to others in all experiments, as shown in the legends of Figures 1, 2, 5, 6.
> >
> > 1. In feature removal and data removal experiments, our methods often achieve superior performance.
> >
> > 1. Furthermore, the feature/data removal metric is not a golden standard, it is just a proxy task. The lack of golden standard is also a `common` issue in the area of interpretatable machine learning. As is also discussed in the second paragraph of Sec. 5.2:
> >  It is hard to tell which ranking is the best because:  1)  There is  no golden standard to determine the true ranking of features;  2) Even if there exists a ground truth ranking of some "perfect model",   the trained xgboost model here might not be able to reproduce it,
> >   since it might not be aligned with the "perfect model''.
> >
> >
> > **Except for the above advantages, we summarize the other  benefits of our approach bellow.**
> >
> >
> > **Benefits compared to classic approaches**
> >
> > 1. Provides a theoretical justification of classical criteria, including Shapley value, Banzhaf value and probabilistic values.  We show that
> > they are all trying to decouple correlations among players under the energy-based framework.
> >
> > 1. Derive a series of new valuation criteria (the $K$-step variational valuations) that all satisfy the minimal set of axioms in order to
> > being a suitable valuation criteria.
> >
> > 1. Flexible tradeoff of decoupling and fairness through tuning the temperature $T$.
> >
> > 1. Enable learnable value functions $F_{\theta} (S)$ when supervision is available, where $\theta$ is the parameter of the value function.
> > For example, for feature interpretation of deep neural nets, $\theta$ could be the
> > parameters of the neural net and the value function $F_{\theta} (S)$
> > being output of the neural net when given features inside $S$ as inputs.
> >
> > 1. Provides a Bayesian framework for valuation problems, under which one can flexibly incorporate priors $P_0(S)$.
> >
> >
> > ### Comment: More justification on the maximum entropy distribution
> >
> > > There are a few ways that the model choices could be further justified.
> > The paper is really built around the specific choice  of the
> > distribution over coalitions, which is justified in a few sentences
> > in the third paragraph of the introduction. However, since the
> > reader really has to buy in to this choice of  to appreciate the rest of the paper, I think that a bit  more justification would be useful.
> >
> >
> > **ANSWER**:  Nice suggestion!
> >
> > The maximum entropy distribution $p(S) \propto \exp(F(S)/T) $ in Eq. (1) is derived from solving the constrained optimization problem in Appendix A.
> >
> > The maximum entropy principle encodes the modeling belief that one shall assume
> > nothing about what is unknown. Another lens to understand it is:
> > Since the distribution with the maximum entropy is the one that makes the fewest assumptions about the true distribution of data, the principle of maximum entropy can be seen as an application of `Occam's razor`.
> > We have elaborated more in the updated appendix.
> >
> > ### Comment: The inverse sigma functions
> > >One other thing I couldn’t quite follow is why the importance score
> > ends up being defined in terms of the inverse sigma functions.
> >
> > **ANSWER**:
> >
> > This is because of the scale of importance score shall be
> > $(- \infty, \infty)$, while the scale of variational marginals $\mathbf{x}$ is $[0, 1]$.
> > So we need to use an inverse sigma function to transform the scale.

---

> > > ### Author Response · Authors · 2021-11-16
> > > **Answers to Reviewer (Part 3)**
> > >
> > >
> > > ### Smaller comments:
> > >
> > > > 3rd paragraph of Section 1: How is mu chosen? Why doesn’t it
> > > show up in Equation (1)?
> > >
> > > **ANSWER**: $\mu$ is used as an assumption of the average payoff in order to
> > > derive the maximum entropy distribution, it also corresponds to the common assumption in statistical mechanics [Jaynes 1957a; b] in the derivation of Maxwell–Boltzmann distribution by Maxwell in 1860 [1,2], where it means the
> > > conservation of average energy in the system.
> > >
> > > It is connnected to the temperature $T$ in an implicit way as shown in
> > > Eq. (25) in the appendix:
> > > $$-  \frac{\partial H_{\text{max}}}{\partial \mu} =: \frac{1}{T}$$
> > >
> > >
> > > [1] Maxwell, J.C. (1860 A): Illustrations of the dynamical theory of gases. Part I. On the motions and collisions of perfectly elastic spheres. The London, Edinburgh, and Dublin Philosophical Magazine and Journal of Science, 4th Series, vol.19, pp.19-32.
> > >
> > > [2] Maxwell, J.C. (1860 B): Illustrations of the dynamical theory of gases. Part II. On the process of diffusion of two or more kinds of moving particles among one another. The London, Edinburgh, and Dublin Philosophical Magazine and Journal of Science, 4th Ser., vol.20, pp.21-37.
> > >
> > >
> > > > 4th paragraph of Section 1: At first, I was confused about what the “valuation problem” is. Then I realized it’s the problem of defining the importance vector; it would be helpful to clarify this.
> > >
> > > **ANSWER**: Nice suggestion! We clarify more on this in the revised manu.
> > >
> > >
> > > > Last paragraph of Section 1: I have a few suggestions:
> > > I didn’t know what was meant by a “decoupling perspective” until I
> > > got to Section 4, so I wasn’t able to understand this sentence.
> > >
> > > **ANSWER**: Good suggestion!
> > >  We have added more explanation in the revised manu.
> > >
> > >
> > > > “Intriguing properties” is a bit too vague, in my opinion.
> > > After reading Section 5, I’m not really sure what these intriguing
> > > properties would be, so it would definitely help to spell these out.
> > >
> > > **ANSWER**:  Intriguing properties mainly  include benefits of our approach.
> > > For example, empirically,
> > > - Lower decoupling error achieved by it compared to all other baselines
> > > - Improved performance for data removal and feature removal experiments.
> > >
> > > Please also refer to the answer to the first question for other theoretical
> > > benefits of our approach.

---

> > > > ### Author Response · Authors · 2021-11-26
> > > > **Further results on scalability of our methods**
> > > >
> > > > We would like to update some new results (which you also asked about) regarding scalability of our methods.
> > > >
> > > > Variational index needs 5~10 iterations to converge. In each iteration one has to evaluate the gradient of multilinear extension  $\nabla f_{\text{mt}}^F(\mathbf{x})$, which needs MCMC sampling to estimate the exponential sum.
> > > >
> > > > Along the discussion with other reviewers,  we come up with a "one-shot sampling trick",  when it is expensive to evaluate the value function  $F(S)$.
> > > > This trick could reuse the sampled values in each iteration,
> > > > __such that Alg. 1 could run with the similar cost as calculating  Banzhaf values__.
> > > >
> > > > The one-shot sampling trick is built upon one formulation taken from Eq. 13:
> > > > $$\nabla_i f_{\text{mt}}^F(\mathbf{x}) =  \sum_{S\subseteq
> > > > 	N - i  }\ [F(S+ i) - F(S)]   q(S; ({\mathbf x | x_{i}\gets 0}))$$
> > > > For coordinate $i$ of $\nabla f_{\text{mt}}^F(\mathbf{x})$,   we can firstly sample $m$ coalitions
> > > > uniformly randomly from $2^{N-i}$, and evaluate their marginal contributions.  Then in each of the following iterations, one could reuse the one-shot sampled
> > > > marginal contributions to estimate the gradient according to the above equation.  In this way, we could make the cost of multi-step running of Alg. 1
> > > > similar as that of Banzhaf value in terms of the number of $F(S)$ evaluations.
> > > > Compared to the original iterative sampling from $q$, this one-shot sampling
> > > > might come with a variance-complexity tradeoff, for which we will explore as a future work.

---

> > > > > ### Author Response · Authors · 2021-11-29
> > > > > **Message to reviewer**
> > > > >
> > > > > Dear Reviewer:
> > > > >
> > > > > Thank you again for your valuable comments! We are wondering if your concerns have been addressed properly. Please let us know if you have any further questions.
> > > > >
> > > > > Best,
> > > > >
> > > > > The authors.

---

### Official Review · Reviewer_Jn7f · 2021-11-02

**Correctness:** 4
**Technical Novelty And Significance:** 4
**Empirical Novelty And Significance:** 3
**Recommendation:** 8
**Confidence:** 3

**Main Review:**

Regarding strengths, to the best of my knowledge the proposed valuation of the framework is novel. It is well motivated and the connections with existing classical methods are very interesting. It also opens the door for further extensions as different surrogate models or application specific priors can be easily incorporated. While it is hard to argue (both in theory and in practice) that one valuation method is better than the alternatives, the empirical results seem to be reasonable. I believe that this paper can have significant impact in the area in the immediate future.

Regarding weaknesses, the paper could be improved in terms of approachability to practitioners. Firstly, reporting the run times of the experiments and/or the accuracy-time trade-offs for MCMC methods would be useful for practitioners. Additionally, any advice on how users should interpret the absolute scores of the Variational Index (especially since $T$ affects their scaling) could be useful. This is especially true for cases where the rankings of different valuation methods are similar but the absolute scores are not. Moreover, the paragraph on why additivity or efficiency does not make sense for some games is very important for practitioners to understand. Practitioners may be easily tricked into thinking that the more properties a valuation measure satisfies the better regardless of the game they try to understand. Right now the paragraph is a bit dense and hard to follow for audiences not familiar with prior work on axiomatization of valuations in cooperative games. Toy numerical examples to demonstrate why additivity or efficiency can result in unintuitive/not-useful valuations could also help practitioners grasp what is the problem.

These weaknesses are minor so I am in favor of accepting this work. I have read the responses of the authors. I find that these additions are going to improve the paper. I thus maintain my score.

**Summary Of The Paper:**

The paper studies valuation problems for cooperative games. It proposes a new valuation measure called Variational Index. The idea is to create a coalition probability distribution based on a maximum entropy criterion. Player valuations are then derived by creating decoupled surrogates of this distribution. The authors then present a gradient ascent algorithm to compute this decoupling. Classical valuation criteria like the Shapley value and the Banzhaf index can be recovered as special cases or modifications of the algorithms iterates.

**Summary Of The Review:**

A well written and novel work on a variational framework for player valuations in cooperative games. While approachability to practitioners could be improved, I am in favor of accepting this work.

---

> ### Author Response · Authors · 2021-11-16
> **Answers to Reviewer (Part 1)**
>
> We thank the reviewer for the extensive comments that help us to improve our work. Please see our answers below.
>
> ### Comment: accuracy-time trade-offs for MCMC methods
> > Firstly, reporting the run times of the experiments and/or the
> accuracy-time trade-offs for MCMC methods would be useful for practitioners.
>
> **ANSWER**:  Thanks for the suggestion!
>
>
> We first analyze the time complexity of estimating the partial derivative
> $\nabla_i f_{\text{mt}}^F(\mathbf{x}^{k})$ using naive MCMC sampling in terms of the number of value function evaluations.  Suppose
> one oracle of evaluating the value function $F(S)$ costs $EO$ time. Sampling
> $m$ coalitions
> from the simple $q$ distribution  is cheap, which can be ignored. Then
> the time complexity of estimating the partial derivative
> $\nabla_i f_{\text{mt}}^F(\mathbf{x}^{k})$ is $O(m \times  EO)$. For the
> full gradient it is $O(n m \times  EO)$.
> If we run Alg. 1 for $K$ epochs, the cost is $O(K n m \times  EO)$.
>
> We indeed observe the effect of different number of samples for the MCMC sampling, which is plotted in Figure 10 of the Appendix in the updated manuscript.
> This is a data valuation problem with 20 players. More experimental results
> will be added once the running is finished.
>
>
> By the way, we would like to note that all game-theoretic criteria have to
> deal with the exponential sum when calculating the gradient of the multilinear
> extension $\nabla f_{\text{mt}}^F(\mathbf{x}^{k})$, whose approximation follows
> from the expression in Eq. 13.
>
>
> ### Comment:
> > Additionally, any advice on how users should interpret the absolute scores of the Variational Index (especially since  affects their scaling) could be useful. This is especially true for cases where the rankings of different valuation methods are similar but the absolute scores are not.
>
> **ANSWER**:  Great suggestion!
>
> Absolute values of the variational index has a clear interpretation in the
> probabilistic framework: variational index (range: $(-\infty, \infty)$) is obtained by transforming the varitional marginals $\mathbf{x}^* \in [0, 1]^n$ by a bijective mapping $\mathbf{s}^*:= T\sigma^{-1}(\mathbf{x}^*)$.
>
> Under the EBM framework, the $i$-th entry of the varitional marginals ${x_i}^*$
> means the odd of the $i$-th player showing up in the coalition in the
> surrogate distribution $q(S; \mathbf{x}^*)$.  So absolute values of the variational index also reflects this odd, but with a different range, e.g.,
> $s_i^* > 0$ corresponding to ${x_i}^* > 0.5$.

---

> > ### Author Response · Authors · 2021-11-16
> > **Answers to Reviewer (Part 2)**
> >
> >
> > ### Comment
> > > Moreover, the paragraph on why additivity or efficiency does not make sense for some games is very important for practitioners to understand. Practitioners may be easily tricked into thinking that the more properties a valuation measure satisfies the better regardless of the game they try to understand. Right now the paragraph is a bit dense and hard to follow for audiences not familiar with prior work on axiomatization of valuations in cooperative games. Toy numerical examples to demonstrate why additivity or efficiency can result in unintuitive/not-useful valuations could also help practitioners grasp what is the problem.
> >
> > **ANSWER**: Great suggestion!
> >
> > By this paragraph we mainly mean satisfying more axioms is not essential for valuation problems. (We have also updated the manu. accordingly)
> >
> > By the bellow example we will try to explain that for valuation problems, satisfying more axioms is not necessary, sometimes even does not make sense. Whether more axioms shall be considered and which sets of them shall be added really depend on the specific scenario being modeled.
> >
> > According to Theorem 1, in general, our proposed $K$-step variational values satisfy the minimal  set of axioms in order to being an appropriate valuation criterion. Note that specific realizations of the $K$-step variational values could also satisfy more axioms, for example,
> > the $1$-step variational value initialied at $0.5*\mathbf{1}$
> >  also satisfies the additivity axiom. Furthermore, we have the following observations:
> >
> > **Satisfying more axioms is not essential for valuation problems.**
> >  Notably, in cooperative game theory, one line of work is to seek for solution concepts that would satisfy more axioms. However, for valuation problems in machine learning, this is arguably not essential.  For example, similar as what [Ridaoui et al 2018] argues,  efficiency   does not make sense for certain games. For a simple illustration,  let us consider a  voting game from a  classification model with 3 binary features $\mathbf{x}\in \{0, 1\}^3$ with weights $\mathbf{w} = [2, 1, 1]^T$:
> > $f(\mathbf{x}) := \text{Indicator}_{\{\mathbf{w}^T \mathbf{x} \geq 3 \}}$.  Now we are trying to find the valuation of each feature in $V = \{x_1, x_2, x_3\}$.
> > Naturally, the value function in the corresponding voting game shall be $F(S) = f(\mathbf{x}_S)$ where $\mathbf{x}_S$ means setting the coordinates of $\mathbf{x}$ inside $S$ to be 1 while leaving others to be 0.
> > In this game let us count  how many times each feature could flip the classification result: for feature $x_1$, there are three situations: $F(\{1,2\}) - F(\{2\})$,   $F(\{1,3\}) - F(\{3 \})$ and
> > $F(\{1,2, 3\}) - F(\{2, 3 \})$; for feature $x_2$, there are one situation: $F(\{1,2\}) - F(\{1\})$; for feature $x_3$, there are one situation: $F(\{1,3\}) - F(\{1\})$. Then the voting power (or valuation) of each feature shall follow a $3:1:1$ ratio.
> > By simple calculations, one can see that the Banzhaf values  of the three features are $\frac{3}{4}, \frac{1}{4}, \frac{1}{4}$, which is consistent with the ratio of the expected voting power. However, the Shapley values of them are $\frac{4}{6}, \frac{1}{6}, \frac{1}{6}$, which is not consistent due  to satisfying the efficiency axiom.

---

> ### Author Response · Authors · 2021-11-30
> **Thanks for reading our responses.**
>
> Dear Reviewer,
>
> Thanks a lot for reading our responses. Your extensive comments have been very valuable for refining the work.
> We will add more details in the final version.
>
> Best,
>
> The authors.

---

### Official Review · Reviewer_28es · 2021-11-06

**Correctness:** 4
**Technical Novelty And Significance:** 4
**Empirical Novelty And Significance:** 4
**Recommendation:** 8
**Confidence:** 4

**Main Review:**

The energy-based perspective is, to my knowledge, a novel perspective for Shapley values and player valuation in ML, and I found it quite cool to see these tools (EBMs, mean-field VI) applied in this way. The variational index is an interesting and it appears to perform well in the experiments relative to Shapley/Banzhaf values.

I have a couple questions/comments that I hope the authors will consider for improving the paper.

**Premise of EBMs for cooperative games.** The idea of using EBMs to analyze cooperative games was a bit confusing and could probably be introduced better. To be specific, the paper starts by saying we should learn a probability distribution over coalitions, but this does not begin to sound like a worthwhile endeavor until several sections in. When we're using Shapley/Banzhaf values, we control the distribution over coalitions/orderings, so learning a distribution sounds (at least initially) like a somewhat pointless idea. Similarly, the idea of finding an entropy-maximizing coalition distribution (constrained to a mean value, which it's not clear how to choose!) does not initially sound useful. Of course it's not pointless, but what ultimately connects these EBM ideas to player valuation is the crucial step of doing mean-field VI. Because of the important role mean-field VI plays here, I wonder if it doesn't deserve a bit more attention and emphasis.

Appendix A attempts to explain this a bit more, but I didn't find these reasons compelling (particularly the second paragraph). Unless I'm missing something, the real reason to use EBMs + mean-field VI is that it enables us to learn a factorized distribution over players that places higher probabilities on players that contribute more value, and this gives us a new perspective for defining player valuations, which happens to connect to existing ideas like Shapley/Banzhaf/probabilistic values.

**Connection with multilinear extensions.** The idea of learning a factorized distribution over players is most similar to the idea of multilinear extensions in cooperative game theory (as in Okhrati and Lipani). In that work, Shapley values are defined as the expected marginal contribution where the preceding coalition is determined by a factorized distribution over players (integrated over a probability value); here, the probabilities of the factorized distribution are learned, but they can still coincide with Shapley values. I wonder if the authors can provide any more commentary on the implicit connection between these ideas, where the probabilities in a factorized distribution can induce player valuations vs. act as player valuations.

**Cooperative game theory in ML.** The paper gives a nice overview of the use of Shapley values in ML, including uses in feature-based explanations, data valuation, and model ensemble valuation. However, a couple key papers are perhaps overlooked and could be cited: Lipovetski & Conklin (2001) and Strumbelj & Kononenko (2010) were some of the first papers to analyze statistical models using Shapley values, and Covert et al. (2020) (already cited) also provides an overview of other papers that use Shapley values, including SPVIM, SAGE and Shapley Effects, and it shows that many other ML explanation methods are also tied to cooperative game theory.

**Entropy gradient.** In section 4.2, the gradient $\nabla_i H(q) = \log \frac{1 - x_i}{x_i}$ is given but the result is not derived. It could be helpful to provide a derivation in the appendix because this result seems non-trivial, and it is important for the subsequent gradient descent routine.

**Gradient descent derivation.** The update rule in algorithm 1 does not immediately look like gradient descent, and I expect it will be confusing to many readers. Where, for example, is the learning rate? I tried to derive this result and if I understand correctly, it comes from taking a gradient step on $\sigma^{-1}(x_i)$ and then applying the sigmoid operation to get $x_i^+$, where the learning rate is chosen as a function of the current value $x_i$. I'm not sure if that's right or if there's a simpler explanation, but there is too much work left to the reader here.

**Practical impact.** I found the ideas in this work very interesting and will view the paper favorably regardless of the answer to this question, but I just wanted to clarify the practical impact. Am I correct in understanding that this energy-based approach does not necessarily offer a more efficient algorithm to calculating Shapley/Banzhaf values? Is the main practical impact, then, proposing the variational index as an alternative to Shapley/Banzhaf values for valuation problems?

**Applying existing Shapley value approximations to the variational index.** In section 3, it's stated that existing Shapley value estimation ideas can be applied directly ("can be seamlessly lifted") to calculating the variational index. That point didn't come up later in the paper and I don't see how that is the case. How, for example, could we use a permutation-based or weighted least squares-based Shapley value estimator to calculate the variational index? Or how could we use a model-specific Shapley value approximation like TreeSHAP? How could these things be integrated into algorithm 1, or be adapted into different routines for optimizing the KL divergence? I don't get this, some clarification on this point would be helpful.

**Role of temperature.** It might be worth noting explicitly for eqs. 14-15 that the specific temperature value does not matter, and that it does not matter for any single-step update; currently, the reader must figure this out for themselves. Aside from that, it's a bit unsatisfactory that different temperatures yield different variational indices, and that we don't know much about the properties of the different solutions, but I suppose it's fine to leave further investigation to future work. It could also be nice to have either a footnote or brief appendix section showing why $T = 0$ and $T \to \infty$ induce even spread and 0/1 probabilities, respectively, as this is also currently left to the reader.

**Role of initializer.** In section 4.2, there's a brief section discussing the initializer's role w.r.t. variational values. It seems mostly right, but I'm confused by the claim that the initializer doesn't matter if you plan on running GD to convergence. How can that be, given that the problem is non-convex (stated earlier in the paper)? These seem like contradictory ideas, please clarify if possible.

**Additivity and efficiency properties.** In section 4.4, there's a paragraph discussing why variational values don't satisfy the additivity and efficiency properties satisfied by Shapley values. I found this paragraph a bit odd: in addressing this "why" question, your explanation addresses why they *shouldn't* (reasons why these properties might be unappealing), as if you had some choice in the matter, rather than the mathematical/mechanistic reasons why they don't. I would ask that you adjust this paragraph to clarify whether you're explaining i) why those properties aren't satisfied or ii) why it's okay that they're not satisfied.

A couple things about the experiments:

- In section 5.3 where we look at the convergence of algorithm 1, we can clearly see that it converges. But does it converge to the same point regardless of the initialization? This may be worth looking into due the problem's non-convexity.
- Is it worth looking into whether algorithm 1 can yield efficient, low-variance Shapley/Banzhaf value estimates relative to existing estimators? Do you have any intuition about how the variance might compare for a fixed number of game evaluations?

There were a couple specific phrases that I thought could be improved:

- The introduction says that you explore a "probabilistic treatment" of games. That's true, but it's not very specific because cooperative game formulations are sometimes probabilistic, there's work considering stochastic cooperative games, and Shapley/Banzhaf values have probabilistic formulations. It might be better to say that you propose learning a factorized distribution over players to arrive at player valuations, because that's what's unique here. The same paragraph says something like this later, but it leaves out the bit about learning a factorized surrogate distribution and the fact that the original distribution is encouraged to put more mass on players that contribute more value.
- Also in the introduction, you state that you "conduct learning and uncertainty analysis in a unified Bayesian manner." I'm not sure this is correct, your method of course does VI, but not uncertainty analysis or Bayesian inference (where's the prior, what's the data?).

**Summary Of The Paper:**

This paper proposes an energy-based perspective on cooperative games that permits a gradient-based calculation of Shapley/Banzhaf values, as well as the definition of a new alternative value - the variational index. A quick summary of the paper's key ideas is:

- For a given cooperative game $F$, we can seek an entropy maximizing distribution over coalitions $p(S)$ that satisfies a constraint on the mean coalition value $\mu$
- Solving the entropy maximization problem via its Lagrangian yields the Boltzmann distribution $p(S) \propto \exp(F(S)/T)$, where the temperature $T$ has a one-to-one correspondence with the mean coalition value $\mu$ (this result is in the appendix). This distribution gives more probability mass to coalitions that achieve higher values
- We can seek a simpler alternative to $p(S)$ by doing mean-field variational inference, i.e., finding a factorized surrogate $q(S)$ where each player's participation is determined by independent Bernoulli RVs. The result will intuitively assign higher probabilities to players that belong to high-value coalitions, so these probabilities can serve a function similar to Shapley/Banzhaf values
- The VI approach suggests a KL divergence minimization (or ELBO maximization) objective for learning $q(S)$, which is parameterized by $x \in [0, 1]^n$. Doing gradient descent on this objective yields a relatively simple update rule, where we repeatedly set $x_i^+ = \sigma(\nabla_i f_{mt}(x) / T)$ for $i = 1, \ldots, n$
- The authors define the "variational index" as a function of the solution to the KL divergence minimization problem: $s^* = T\sigma^{-1}(x^*)$
- The authors find that the Banzhaf value can be found using a single-step update to a particular initialization of the KL divergence minimization problem (luckily the temperature $T$ is not important for single-step updates). Similarly, they find that the Shapley value is the average of the single-step update applied to different initializations (again, the temperature doesn't matter). Finally, the authors point out that any single-step update applied to a symmetric initialization will be a probabilistic value (a class of solution concepts in cooperative game theory, of which Shapley/Banzhaf values are special cases)
- Lastly, the authors suggest a practical sampling-based approach to calculating the necessary gradients, which are just as difficult to calculate as the Shapley/Banzhaf values because they require calculating the value for every coalition $S \subseteq N$

The experiments compare the variational index to Shapley and Banzhaf values in data and feature removal tasks, finding that it performs quite favorably in the settings examined.

**Summary Of The Review:**

This paper introduces some very interesting ideas about the use of EBMs and mean-field VI in the context of player valuation for cooperative games. Their new perspective is connected to Shapley/Banzhaf/probabilistic values, it permits approximate optimization (the gradients require sampling), and they define a new value (the variational index) that performs quite well in their experiments.

I had a couple questions and comments about the writing, but I expect these will be easy to address.

---

> ### Author Response · Authors · 2021-11-16
> **Answers to Reviewer (Part 1)**
>
> We thank the reviewer for the extensive comments that help us to improve our work. Please see our detailed answers below.
>
>
> ### Comment: **Premise of EBMs for cooperative games.**
> > The idea of using EBMs to analyze cooperative games was a bit confusing and could probably be introduced better.
> >To be specific, the paper starts by saying we should learn a probability distribution over coalitions, but this does not begin to sound like a worthwhile endeavor until several sections in. When we're using Shapley/Banzhaf values, we control the distribution over coalitions/orderings, so learning a distribution sounds (at least initially) like a somewhat pointless idea.
> >Similarly, the idea of finding an entropy-maximizing coalition distribution (constrained to a mean value, which it's not clear how to choose!) does not initially sound useful. Of course it's not pointless, but what ultimately connects these EBM ideas to player valuation is the crucial step of doing mean-field VI. Because of the important role mean-field VI plays here, I
> wonder if it doesn't deserve a bit more attention and emphasis.
> > Appendix A attempts to explain this a bit more, but I didn't find these reasons compelling (particularly the second paragraph). Unless I'm missing something, the real reason to use EBMs + mean-field VI is that it enables us to learn a factorized distribution over players that places higher probabilities on players that contribute more value, and this gives us a new perspective for defining player valuations, which happens to connect to existing ideas like Shapley/Banzhaf/probabilistic values.
>
>
> **ANSWER**:  Great suggestion!  We agree that the main benefits of adopting the
> maximum entropy framework show up a bit later, we provide more explanations
> regarding the maximum entropy distribution in Appendix A.
>
> Conducting mean-field VI is the crucial step to connect with the
> classical criteria, at the same time, introducing the new $K$-step
> variational valuations.  However, without introducing the EBM framework,
> one cannot do the mean-field VI. We have changed the writting to emphasis more on the role of mean-field VI.
>
> Another benefit of introducing a maximum entropy distribution is to enable
> learning $F(S)$ in a principled manner, which is not the focus of
> this work. Specifically,  it enables learnable value functions $F_{\theta} (S)$ when supervision is available, where $\theta$ is the parameter of the value function.
> For example, for feature interpretation of deep neural nets, $\theta$ could be the
> parameters of the neural net and the value function $F_{\theta} (S)$
> being output of the neural net when given features inside $S$ as inputs.
>
> **Regarding how to choose $\mu$:** $\mu$ is used as an assumption of the average payoff in order to
> derive the maximum entropy distribution. It also corresponds to the common assumption in statistical mechanics [Jaynes 1957a; b] in the derivation of Maxwell–Boltzmann distribution by Maxwell in 1860 [1,2], where it means the
> conservation of average energy in the system.
>
> It is connected to the temperature $T$ in an implicit way as shown in
> Eq. (25) in the appendix:
> $$-  \frac{\partial H_{\text{max}}}{\partial \mu} =: \frac{1}{T}.$$
>
>
> [1] Maxwell, J.C. (1860 A): Illustrations of the dynamical theory of gases. Part I. On the motions and collisions of perfectly elastic spheres. The London, Edinburgh, and Dublin Philosophical Magazine and Journal of Science, 4th Series, vol.19, pp.19-32.
>
> [2] Maxwell, J.C. (1860 B): Illustrations of the dynamical theory of gases. Part II. On the process of diffusion of two or more kinds of moving particles among one another. The London, Edinburgh, and Dublin Philosophical Magazine and Journal of Science, 4th Ser., vol.20, pp.21-37.

---

> > ### Author Response · Authors · 2021-11-16
> > **Answers to Reviewer (Part 2)**
> >
> > ### Comment: **Connection with multilinear extensions.**
> > >The idea of learning a factorized distribution over players is most similar to the idea of multilinear extensions in cooperative game theory (as in Okhrati and Lipani). In that work, Shapley values are defined as the expected marginal contribution where the preceding coalition is determined by a factorized distribution over players (integrated over a probability value); here, the probabilities of the factorized distribution are learned, but they can still coincide with Shapley values.
> > >I wonder if the authors can provide any more commentary on the implicit connection between these ideas, where the probabilities in a factorized distribution can induce player valuations vs. act as player valuations.
> >
> > **ANSWER** Nice comments! There indeed exist implicit connections with works on multilinear extensions, as we noted when deriving the multilinear extensions
> > in Sec. 4.1. Different to prior works of using multilinear extensions
> > as a representation,  our work here provides an approach to
> > derive the multilinear extension through the mean-field VI framework under the maximum entropy distribution, which also verifies the rationality of multilinear extension of cooperative games.
> >
> > ### Comment: **Cooperative game theory in ML.**
> > >The paper gives a nice overview of the use of Shapley values in ML, including uses in feature-based explanations, data valuation, and model ensemble valuation. However, a couple key papers are perhaps overlooked and could be cited: Lipovetski & Conklin (2001) and Strumbelj & Kononenko (2010) were some of the first papers to analyze statistical models using Shapley values, and Covert et al. (2020) (already cited) also provides an overview of other papers that use Shapley values, including SPVIM, SAGE and Shapley Effects, and it shows that many other ML explanation methods are also tied to cooperative game theory.
> >
> >
> > **ANSWER**: Great suggestion! We have updated the manuscript  to include more
> > relevant references on Shapley values in ML.
> >
> >
> > ### Comment: **Entropy gradient**.
> > >In section 4.2, the gradient
> >  is given but the result is not derived. It could be helpful to provide a derivation in the appendix because this result seems non-trivial, and it is important for the subsequent gradient descent routine.
> >
> > **ANSWER**: Nice suggestion! We add the derivation in  Appendix F of the revised manuscript.
> >
> >
> >
> >
> > ### Comment: **Gradient descent derivation.**
> > > The update rule in algorithm 1 does not immediately look like gradient descent, and I expect it will be confusing to many readers. Where, for example, is the learning rate? I tried to derive this result and if I understand correctly, it comes from taking a gradient step on
> >  and then applying the sigmoid operation to get
> > , where the learning rate is chosen as a function of the current value
> > . I'm not sure if that's right or if there's a simpler explanation, but there is too much work left to the reader here.
> >
> >
> > **ANSWER**:  Thanks a lot for pointing this out! A more appropriate
> > way to call it shall be  "fixed point iteration", we have changed the phrase in the revised version. A side note on this point is that:  The fixed point iteration actually
> > corresponds to gradient ascent with adaptive step size vector $\alpha^k$ as,
> > $$
> > \alpha_i^k = \frac{\sigma(\nabla_i f_{\text{mt}}^F(\mathbf{x}^{k})/T) - {x}_i^k }{\nabla_i f_{\text{mt}}^F(\mathbf{x}^{k})/T + \log \frac{1-{x}_i^k}{{x}_i^k} }
> > $$

---

> > > ### Author Response · Authors · 2021-11-16
> > > **Answers to Reviewer (Part 3)**
> > >
> > >
> > > ### Comment: **Practical impact.**
> > > > I found the ideas in this work very interesting and will view the paper favorably regardless of the answer to this question, but I just wanted to clarify the practical impact. Am I correct in understanding that this energy-based approach does not necessarily offer a more efficient algorithm to calculating Shapley/Banzhaf values? Is the main practical impact, then, proposing the variational index as an alternative to Shapley/Banzhaf values for valuation problems?
> > >
> > > **ANSWER**: Great comment! We agree that the proposed variational
> > > values also suffer from exponential computational complexity as classical
> > > criteria.
> > >
> > > Meanwhile, we also would like to highlight other practical benefits:
> > >
> > > 1. Decoupling error (see Def. 1) is a clear performance metric to measure the decoupling
> > > performance.  The proposed variational index always enjoys lower decoupling error compared to others in all experiments.
> > >
> > > 1. In feature removal and data removal experiments, our methods often
> > > achieve superior performance.
> > >
> > > 1. Notably, the feature/data removal metric is not a golden standard, it
> > > is just a proxy task. The lack of golden standard is also a `common` issue in the area of interpretable machine learning. As is also discussed in the second paragraph of Sec. 5.2:  It is hard to tell which ranking is the best because:  1)  There is  no golden standard to determine the true ranking of features;  2) Even if there exists a ground truth ranking of some "perfect model", the trained xgboost model here might not be able to reproduce it,
> > >   since it might not be aligned with the "perfect model''.
> > >
> > >
> > >
> > > ### Comment: **Applying existing Shapley value approximations to the variational index.**
> > > > In  section 3, it's stated that existing Shapley value estimation ideas can be applied directly ("can be seamlessly lifted") to calculating the variational index. That point didn't come up later in the paper and I don't see how that is the case. How, for example, could we use a permutation-based or weighted least squares-based Shapley value estimator to calculate the variational index? Or how could we use a model-specific Shapley value approximation like TreeSHAP? How could these things be integrated into algorithm 1, or be adapted into different routines for optimizing the KL divergence? I don't get this, some clarification on this point would be helpful.
> > >
> > >
> > > **ANSWER**:  Great question!
> > >
> > > We were  mainly trying to draw the connection between the proposed variational values, Shapley
> > > and Banzhaf values. The calculation of all of them is related to approximating the gradient of
> > > multilinear extension  $\nabla f_{\text{mt}}^F(\mathbf{x}^{k})$.
> > >
> > > The statement in Section 3 is not accurate, we have corrected the statement in the revised paper.
> > >
> > >
> > > ### Comment: **Role of temperature.**
> > > > It might be worth noting explicitly for eqs. 14-15 that the specific temperature value does not matter, and that it does not matter for any single-step update; currently, the reader must figure this out for themselves. Aside from that, it's a bit unsatisfactory that different temperatures yield different variational indices, and that we don't know much about the properties of the different solutions, but I suppose it's fine to leave further investigation to future work.
> > > It could also be nice to have either a footnote or brief appendix section showing why $T \rightarrow \infty$  and $T=0$  induce even spread and 0/1 probabilities, respectively, as this is also currently left to the reader.
> > >
> > >
> > > **ANSWER**:
> > >
> > > Great to point out that Eqs 14, 15 do not depend on the temperature.
> > > The temperature $T$ makes a difference only when one runs Alg. 1 for more
> > > than one iterations.
> > >
> > >
> > > **The extreme situations  of  $T \rightarrow \infty$  and $T=0$**:  This
> > > is a classical phonenmon  regarding the role of temperature in Boltzman distributions. Here we give an
> > > intuitive  explanation based on the fixed point iteration $\mathbf{x}^{k+1} = \sigma(\nabla f_{\text{mt}}^F(\mathbf{x}^{k})/T)$.
> > >
> > > Assume $|F(S+i) - F(S)| < \infty, \forall i, S$. So $|\nabla f_{\text{mt}}^F(\mathbf{x}^{k})| < \infty$.
> > >
> > > Here when $T \rightarrow \infty$, $\nabla f_{\text{mt}}^F(\mathbf{x}^{k})/T \rightarrow 0$, after applying the sigmoid operation, $\mathbf{x}^{k+1} \rightarrow 0.5\times \mathbf{1}$, so we reach a uniform/spread/flat valuation.
> > >
> > > When $T \rightarrow 0$, $\nabla f_{\text{mt}}^F(\mathbf{x}^{k})/T$ becomes either $\infty$ or $-\infty$. After applying the sigmoid operation, $\mathbf{x}^{k+1}$ becomes either 1 or 0, so one gets a sharp (or 0/1) distribution.

---

> > > > ### Author Response · Authors · 2021-11-16
> > > > **Answers to Reviewer (Part 4)**
> > > >
> > > >
> > > > ### Comment: **Role of initializer.**
> > > >
> > > > > In section 4.2, there's a brief section discussing the initializer's
> > > > role w.r.t. variational values. It seems mostly right, but I'm confused by the claim that the initializer doesn't matter if you plan on running GD to convergence. How can that be, given that the problem is non-convex (stated earlier in the paper)? These seem like contradictory ideas, please clarify
> > > >  if possible.
> > > >
> > > > **ANSWER**: Thanks for pointing this out! The EBLO objective is indeed non-concave/non-convex in general.  So if we run the fixed point iteration for
> > > > many steps, it will converge to stationary points of ELBO objective.
> > > > This is what we mainly meant in the discussion.  We have refined the discussion
> > > > in the revised version to avoid potential ambiguity.
> > > >
> > > >
> > > >
> > > > ### Comment: **Additivity and efficiency properties.**
> > > >
> > > > > In section 4.4, there's a paragraph discussing why variational values don't satisfy the additivity and efficiency properties satisfied by Shapley values. I found this paragraph a bit odd: in addressing this "why" question, your explanation addresses why they shouldn't (reasons why these properties might be unappealing), as if you had some choice in the matter, rather than the mathematical/mechanistic reasons why they don't.
> > > > I would ask that you adjust this paragraph to clarify whether you're explaining i) why those properties aren't satisfied or ii) why it's okay that they're not satisfied.
> > > >
> > > >
> > > > **ANSWER**: Great question! We mainly would like to explain why
> > > > it is okay that these properties are not satisfied.
> > > >
> > > > We have adjusted the paragraph as bellow. By the bellow example we are trying to explain that for valuation problems, satisfying more axioms is not necessary, sometimes even does not make sense. Whether more axioms shall be considered and which sets of them shall be added really depend on the specific scenario being modeled, which will be left for important future work.
> > > >
> > > >
> > > > According to Theorem 1, in general,  our proposed $K$-step variational values satisfy the minimal  set of axioms in order to being a suitable valuation criterion.
> > > > Note that specific realizations of the $K$-step variational values could also satisfy more axioms, for example,
> > > > the $1$-step variational value initialied at $0.5*\mathbf{1}$
> > > >  also satisfies the additivity axiom. Furthermore, we have the following observations:
> > > >
> > > > **Satisfying more axioms is not essential for valuation problems.** Notably, in cooperative game theory, one line of work is to seek for solution concepts that would satisfy more axioms.
> > > > However, for valuation problems in machine learning, this is arguably not essential.  For example, similar as what [Ridaoui et al 2018] argues,  efficiency   does not make sense for certain games. For a simple illustration,  let us consider a  voting game from a  classification model with 3 binary features $\mathbf{x}\in \{0, 1\}^3$ with weights $\mathbf{w} = [2, 1, 1]^T$:
> > > > $f(\mathbf{x}) := \text{Indicator}_{\{\mathbf{w}^T \mathbf{x} \geq 3 \}}$.  Now we are trying to find the valuation of each feature in $V = \{x_1, x_2, x_3\}$.
> > > > Naturally, the value function in the corresponding voting game shall be $F(S) = f(\mathbf{x}_S)$ where $\mathbf{x}_S$ means setting the coordinates of $\mathbf{x}$ inside $S$ to be 1 while leaving others to be 0.
> > > > In this game let us count  how many times each feature could flip the classification result: for feature $x_1$, there are three situations: $F(\{1,2\}) - F(\{2\})$,   $F(\{1,3\}) - F(\{3 \})$ and
> > > > $F(\{1,2, 3\}) - F(\{2, 3 \})$; for feature $x_2$, there are one situation: $F(\{1,2\}) - F(\{1\})$; for feature $x_3$, there are one situation: $F(\{1,3\}) - F(\{1\})$. Then the voting power (or valuation) of each feature shall follows a $3:1:1$ ratio.
> > > > By simple calculations, one can see that the Banzhaf indices of the three features are $\frac{3}{4}, \frac{1}{4}, \frac{1}{4}$, which is consistent with the ratio of the expected voting power. However, the Shapley values of them are $\frac{4}{6}, \frac{1}{6}, \frac{1}{6}$, which is not consistent due  to satisfying the efficiency axiom.

---

> > > > > ### Author Response · Authors · 2021-11-16
> > > > > **Answers to Reviewer (Part 5)**
> > > > >
> > > > >
> > > > > ### Comment:
> > > > > > In section 5.3 where we look at the convergence of algorithm 1, we can
> > > > > clearly see that it converges. But does it converge to the same point regardless of the initialization? This may be worth looking into due the problem's non-convexity.
> > > > > Is it worth looking into whether algorithm 1 can yield efficient, low-variance Shapley/Banzhaf value estimates relative to existing estimators? Do you have any intuition about how the variance might compare for a fixed number of game evaluations?
> > > > >
> > > > > **ANSWER**: Nice comments!
> > > > >
> > > > > Alg. 1 will converge to the stationary points of the EBLO objective.
> > > > > In general, multiple stationary points could exist. For some class
> > > > > of games (set functions $F(S)$) whose multilinear extensions are related to monotone operators,
> > > > > there is a single stationary point, and the fixed point iteration would
> > > > > converge to it, as shown by [Ryu, Ernest K., and Stephen Boyd. "Primer on monotone operator methods." Appl. Comput. Math 15, no. 1 (2016): 3-43.].  This will be left for interesting future work.
> > > > >
> > > > > Alg. 1 defines a series of variational valuations through the fixed point
> > > > > iteration, each step crucially depends on the gradient of the multilinear extension. So a better sampling algorithm than MCMC to estimate gradient of the multilinear extension would help with approximating all the valuation criteria. It is noteworthy
> > > > > that the sampling algorithm based on the multilinear form of Shapley value is shown to have lower variance compared to the permutation based sampling algorithm [Okhrati & Lipani 2020], given the same number of game evaluations.
> > > > >
> > > > >
> > > > > ### Comment:
> > > > > > The introduction says that you explore a "probabilistic treatment" of games. That's true, but it's not very specific because cooperative game formulations are sometimes probabilistic, there's work considering stochastic cooperative games, and Shapley/Banzhaf values have probabilistic formulations. It might be better to say that you propose learning a factorized distribution over players to arrive at player valuations, because that's what's unique here. The same paragraph says something like this later, but it leaves out the bit about learning a factorized surrogate distribution and the fact that the original distribution is encouraged to put more mass on players that contribute more value.
> > > > > Also in the introduction, you state that you "conduct learning and uncertainty analysis in a unified Bayesian manner." I'm not sure this is correct, your method of course does VI, but not uncertainty analysis or Bayesian inference (where's the prior, what's the data?).
> > > > >
> > > > >
> > > > > **ANSWER**:  Nice comment!
> > > > >
> > > > > We agree that a "probabilistic treatment" is a bit too general. The maximum
> > > > > entropy treatment would be more appropriate, we have also theoretically
> > > > > verified its specific formulation.
> > > > > The factorized surrogate distribution  $q$ also depends on the original
> > > > > maximum entropy distribution $p(S)$, cause otherwise the "surrogate" would
> > > > > not make sense.
> > > > >
> > > > > Currently we have not considered prior or uncertainty analysis, though
> > > > > the framework can be adapted to a fully Bayesian manner by considering some
> > > > > priors in the Boltzman formulation.  We have
> > > > > modified the writting to get rid of the Bayesian part so far.
> > > > >
> > > > > Regarding learning, we mean the situation where $F(S)$ involves some learnable
> > > > > parameters $\theta$.  It enables learnable value functions $F_{\theta} (S)$ when supervision is available, where $\theta$ is the parameter of the value function.
> > > > > For example, for feature interpretation of deep neural nets, $\theta$ could be the
> > > > > parameters of the neural net and the value function $F_{\theta} (S)$
> > > > > being output of the neural net when given features inside $S$ as inputs.

---

> > ### Comment · Reviewer_28es · 2021-11-20
> > **Thanks for detailed response**
> >
> > Thanks for reading all the reviewer's comments closely and being open to making adjustments. Your responses were very thorough, and I think the various changes discussed here will help the paper quite a bit, making it more interesting and easier for readers to follow.
> >
> > There are a couple things I might emphasize, but at this point it's up to the authors because I'll probably keep my score as is.
> >
> > **About the maximum entropy distribution.** I get that this has some precedent in statistical mechanics, but the justification that it assumes as little as possible about the data distribution (an explanation the authors provided in another response) is not very compelling to me. The way I see it, we don't need such an abstract justification: this is the distribution that yields connections with Shapley and Banzhaf values, and that alone makes it worth using. (We could ask **why** it's the entropy maximizing distribution that yields these connections, but I don't think the paper actually answers that question.)
> >
> > Also, a more intuitive explanation for this distribution is that it turns the game into a probability distribution while assigning probability mass in proportion to the coalitions' values. It's interesting that it can be derived from an entropy maximization perspective, but this aspect seems more important, because it's what makes the variational marginals meaningful as credit allocations when we perform mean-field VI.
> >
> > **About the practical impact.** Thanks for the clarification in your response. On the point about the decoupling error, I think it's a minor misconception that we don't have comparable metrics that might make the Shapley or Banzhaf values optimal in some sense; for example, both values can be derived as solutions to weighted least squares problems (see KernelSHAP for the Shapley value), and the loss value in these problems provides an alternative to the decoupling error where the variational index would appear inferior. In other words, all three values have their own unique optimization-based/variational characterizations. So it's nice to have the decoupling error, but there's now a subtle theoretical question of which metric is better in each scenario, and I'm not sure there's a clear answer.
> >
> > About the performance metrics in the feature/data removal experiments: I agree that these results are quite promising.
> >
> > **The introduction.** I think the slight adjustments to the introduction are an improvement. A couple small things to mention are:
> > - I don't get how the EBM + VI perspective makes it any easier to perform learning on the game $F$ than alternative valuation approaches. As you said in your response to another reviewer, this isn't explored in the paper, and I'm not asking you to run any experiments, but it seems like an overreach to say this is now doable with these new tools. For example, say you want to learn $F$ with an objective that incorporates the Shapley/Banzhaf values or variational index. Previously, it would have been very difficult to do so because you would need to do autograd through a very memory- and compute-intensive estimation procedure. With this new perspective, you have the same problem - estimating the Shapley/Banzhaf values or variational index is not any easier now. So unless I'm missing something, I don't get how this would make it any easier to learn $F$.
> > - As described above, I don't find the current justification around the entropy-maximizing distribution very intuitive. An alternative way of writing the introduction might be to say something like: "We adopt an EBM formulation of the cooperative game, creating a probability distribution where probability mass is allocated in proportion to the coalitions' values. This is equivalent to finding the entropy-maximizing distribution, but more importantly, it enables us to perform mean-field VI and learn probability parameters that function as player valuations." This would reflect that the EBM formulation is necessary/useful because of what it enables, not because entropy maximization is important on its own here (something that 5RYD alluded to as well). Up to you, though.

---

> > > ### Author Response · Authors · 2021-11-20
> > > **Answer to further comments (part 1)**
> > >
> > >
> > > Thanks a lot for your futher comments! Your suggestions are very
> > > helpful for further improving the work, and we are refining the manuscript accordingly. Our specific answers are listed below:
> > >
> > > > **About the maximum entropy distribution.**  I get that this has some precedent in statistical mechanics, but the justification that it assumes as little as possible about the data distribution (an explanation the authors provided in another response) is not very compelling to me.
> > > The way I see it, we don't need such an abstract justification: this is the distribution that yields connections with Shapley and Banzhaf values, and that alone makes it worth using. (We could ask **why** it's the entropy maximizing distribution that yields these connections, but I don't think the paper actually answers that question.)
> > >
> > > >Also, a more intuitive explanation for this distribution is that it turns the game into a probability distribution while assigning probability mass in proportion to the coalitions' values. It's interesting that it can be derived from an entropy maximization perspective, but this aspect seems more important, because it's what makes the variational marginals meaningful as credit allocations when we perform mean-field VI.
> > >
> > > **ANSWER:**
> > >
> > > We agree that adding "finding the distribution that yields connections with Shapley and Banzhaf values" as  the motivation would make it more intuitive.  Meanwhile, justification on the maximum entropy priciple from a minimum prior knowledge perspective will provide a sound foundation of the above connections. So we tend to discuss both aspects as motivations in the refined manu.
> > >
> > >
> > > > **About the practical impact.** Thanks for the clarification in your response. On the point about the decoupling error, I think it's a minor misconception that we don't have comparable metrics that might make the Shapley or Banzhaf values optimal in some sense; for example, both values can be derived as solutions to weighted least squares problems (see KernelSHAP for the Shapley value), and the loss value in these problems provides an alternative to the decoupling error where the variational index would appear inferior. In other words, all three values have their own unique optimization-based/variational characterizations. So it's nice to have the decoupling error, but there's now a subtle theoretical question of which metric is better in each scenario, and I'm not sure there's a clear answer.
> > >
> > > > About the performance metrics in the feature/data removal experiments: I agree that these results are quite promising.
> > >
> > >
> > > **ANSWER:**
> > >
> > > We agree that which metric is better in each scenario is unclear theoretically.
> > > Our preliminary understandings along this open direction are:
> > >
> > > - The construction of the least squares objective is   based
> > > on the formulation of classical game-theoretic criteria, e.g., for the Shapley
> > > value, one uses Shapley kernel and for the Banzhaf value, one uses Banzhaf kernel. In this way, the solution to the least square problem would exactly
> > > recover the corresponding values.
> > >
> > > - The construction of decoupling error is based on the mean-field VI framework, i.e., based on solving the ELBO maximization problem.
> > >
> > > So probably we can call all of them  "criterion-induced" metrics, that is,
> > > we firstly have some criterion and the way to calculate it in mind, and then define a metric based on how much the solution is deviated from the optimal solution.
> > >
> > > Based on the above discussion, we think that a nice start to develop a
> > > "golden" metric would be to find some "criterion-independent" metric, which
> > > at the same time, shall admit precise quantifications.  Along this line of
> > > pursuit, the valuation performance of valuation problems might be a good
> > > direction, as long as one can find some "golden" ground truth, for example,
> > > for the data/feature valuation problems.

---

> > > > ### Author Response · Authors · 2021-11-20
> > > > **Answer to further comments (part 2)**
> > > >
> > > >
> > > > > **The introduction**. I think the slight adjustments to the introduction are an improvement. A couple small things to mention are:
> > > >
> > > > >I don't get how the EBM + VI perspective makes it any easier to perform learning on the game $F$ than alternative valuation approaches. As you said in your response to another reviewer, this isn't explored in the paper, and I'm not asking you to run any experiments, but it seems like an overreach to say this is now doable with these new tools. For example, say you want to learn  with an objective that incorporates the Shapley/Banzhaf values or variational index. Previously, it would have been very difficult to do so because you would need to do autograd through a very memory- and compute-intensive estimation procedure. With this new perspective, you have the same problem - estimating the Shapley/Banzhaf values or variational index is not any easier now. So unless I'm missing something, I don't get how this would make it any easier to learn $F$.
> > > >
> > > > > As described above, I don't find the current justification around the entropy-maximizing distribution very intuitive. An alternative way of writing the introduction might be to say something like: "We adopt an EBM formulation of the cooperative game, creating a probability distribution where probability mass is allocated in proportion to the coalitions' values. This is equivalent to finding the entropy-maximizing distribution, but more importantly, it enables us to perform mean-field VI and learn probability parameters that function as player valuations." This would reflect that the EBM formulation is necessary/useful because of what it enables, not because entropy maximization is important on its own here (something that 5RYD alluded to as well). Up to you, though.
> > > >
> > > > **ANSWER:**
> > > >
> > > > Thanks for the comments! We agree that your suggestions will make the introduction  more intuitive and focused, and we are refining the manu. accordingly.
> > > >
> > > > In terms of learning $F_\theta(S)$, we agree
> > > > that it is a bit too early to discuss them in this work, and its advantage
> > > > does not lie in being more computational efficient. As a pure
> > > > discussion on this research direction, though, we would like to clarify the main advantages of EBM-based treatment for  learning $F_\theta(S)$ here:
> > > >
> > > > - One can apply efficient training techniques of EBMs (potentially with
> > > > some adaptions), such as the  contrastive divergence [Hinton, 2002], noise contrastive estimation [Gutmann and Hyvärinen, 2010] and score matching [Hyvärinen, 2005].
> > > >
> > > > - The mean-field VI would output marginals $\mathbf{x}\in [0, 1]^n$
> > > > of the surrogate distribution, which has an interpretation as the odd
> > > > of players showing up. This may be more appropriate to be used in composing
> > > >  losses when supervisions are available, e.g. in the form of selected
> > > > subset of players (cross entropy loss) or ranking of players. On a high
> > > > level, the similar idea has also been explored heavily in semantic segmentation problems, such as [1] and its followup works.
> > > >
> > > > [1] Zheng, Shuai, Sadeep Jayasumana, Bernardino Romera-Paredes, Vibhav Vineet, Zhizhong Su, Dalong Du, Chang Huang, and Philip HS Torr. "Conditional random fields as recurrent neural networks." In Proceedings of the IEEE international conference on computer vision, pp. 1529-1537. 2015.

---

### Official Review · Reviewer_LgG5 · 2021-11-10

**Correctness:** 4
**Technical Novelty And Significance:** 4
**Empirical Novelty And Significance:** 2
**Recommendation:** 6
**Confidence:** 3

**Main Review:**

P. S. I'm somewhat of an outsider to this topic, with minimal familiarity with cooperative games.

Pros:
- I find the observation that classical criteria such as Shapely value can be seen as one-step "decoupling" approximation to the maximum entropy probabilistic assignment to be quite interesting. In hindsight this seems like a natural connection, although I'm not sure if an expert on this topic would be equally excited about this result.
- The multi-step approximation criterion that the authors propose also seems like a natural extension to existing game-theoretic criteria

Cons:
- I don't find the experimental results to be compelling. They do not convincingly demonstrate that the proposed new criterion has some additional value over existing criteria for ML applications. I understand that the authors have chosen to make the mean-field perspective the central focus of the paper, but I think its important to also have a strong empirical section showcasing utility to an ML audience.
- The feature selection experiments in Sec 5.2 show some improvements over two classical game-theoretic criteria, but the field of subset selection is now quite mature (e.g. https://arxiv.org/pdf/2006.15412.pdf), and comparing to just the two closest game-theoretic criteria doesn't seem to make a strong empirical case.
- The new criterion comes with the added cost of Monte Carlo sampling to approximate exponential sums over multiple rounds, but I'm not entirely sure if the experiments justify this added cost needed.
- The writing is accessible, but can be improved to better motivate the subset/feature selection applications, especially for an ML audience, and provide more elaborate intuition early on for the "decoupling" approximations that the classical criteria seek to compute.

Other comments/questions:
- Alg 1: You refer to this algorithm as performing gradient ascent. To me it appears more like a fixed point iteration algorithm, and not necessarily performing a gradient-based ascent step on the ELBO objective. Please correct me if I'm missing something here.
- Sec 5: Might be good to mention how the decoupling error is computed in all your experiments, given that it requires calculating (an approximation of) the Boltzmann distribution.
- Sec 5.1: I didn't quite understand the role of the data clustering in the experiment. On the x-axis in Fig 1, do you add clusters of data points instead of individual data points?
- Sec 5.2: In Fig 2, you plot the predicted probabilities as you drop features. Is this the average probability predicted by the model across all test examples? Might be good to elaborate why this is a good evaluation metric to look at.
- Sec 1/Intro: I think your mention of a solution concept \phi(F) needs elaboration for audience not necessarily familiar with cooperative games. Similarly, in Sec 2, it wasn't initially clear to me how the importance weights you write out in (2) and (3) relate to the solution concept \phi(F).

**Summary Of The Paper:**

Valuation criteria based on game-theory (e.g. Shapely value) have been used in the ML literature for analyzing feature importance and for data subset selection. These criteria serve as solution concepts for cooperative games and have been adapted by some works in ML for subset valuation problems.

The present paper presents a probabilistic treatment of cooperative games, and shows that two classical valuation criteria can be seen as a one-step factored approximation to maximum entropy solution to the game. They then propose a new valuation criterion "Variational Index" that uses a multi-step factored approximation and show it satisfies some common axioms for cooperative games. The paper also has experimental results on the proposed criterion.



**Summary Of The Review:**

The paper presents some interesting theoretical connections, but does not provide sufficiently compelling empirical results showcasing utility for applications in ML.

---

> ### Author Response · Authors · 2021-11-16
> **Answers to Reviewer**
>
>
> We thank the reviewer for the extensive comments that help us to improve our work.
> It is noteworthy that there are several misunderstandings regarding
> this work,  such as the problem setting being studied, benefits of our approach,
> properties of baselines and so on. Please see our detailed clarifications below.
>
>
> ### Comments: Empirical results and benefits
>
> > I don't find the experimental results to be compelling. They do not convincingly demonstrate that the proposed new criterion has some
> additional value over existing criteria for ML applications. I
> understand that the authors have chosen to make the mean-field
> perspective the central focus of the paper, but I think its important
> to also have a strong empirical section showcasing utility to an
> ML audience.
>
>
> **ANSWER**: Thanks for the comments!   There are misunderstandings regarding the empirical and theoretical
> benefits of our approach. For which we would like to clarify bellow:
>
>
> **1. Empirical benefits.**
>
> 1. Decoupling error (see Def. 1) is a clear performance metric to measure the decoupling performance.  The proposed variational index always enjoys lower decoupling error compared to other baselines in all experiments, as shown in the legends of Figures 1, 2, 5, 6.
>
> 1. In feature removal and data removal experiments, our methods often
> achieve superior performance.
>
> 1. Notably, the feature/data removal metric is not a golden standard, it
> is just a proxy task for measuring valuation performance. The lack of golden standard is also a `common` issue in the area of interpretable machine learning. As is also discussed in the second paragraph of Sec. 5.2:
>  It is hard to tell which ranking is the best because:  1)  There is
>  no golden standard to determine the true ranking of features;  2) Even if there exists a ground truth ranking of some "perfect model",  the trained xgboost model here might not be able to reproduce it,
>   since it might not be aligned with the "perfect model''.
>
>
> **2. Our method also enjoys other benefits compared to classic approaches**
>
> 1. Provides a theoretical justification of classical criteria, including Shapley value, Banzhaf value and probabilistic values.  We show that
> they are all trying to decouple correlations among players under the energy-based framework.
>
> 1. Derive a series of new valuation criteria (the $K$-step variational
> valuations) that all satisfy the minimal set of axioms in order to
> being an appropriate valuation criterion.
>
> 1. Flexible tradeoff of decoupling and fairness through tuning the
> temperature $T$.
>
> 1. Enable learnable value functions $F_{\theta} (S)$ when supervision is available, where $\theta$ is the parameter of the value function.
> For example, for feature interpretation of deep neural nets, $\theta$ could be the
> parameters of the neural net and the value function $F_{\theta} (S)$
> being output of the neural net when given features inside $S$ as inputs.
>
> 1. Provides a Bayesian framework for valuation problems, under which
> one can flexibly incorporate priors $P_0(S)$.
>
>
>
> ### Comment: feature subset selection vs valuation problems
> > The feature selection experiments in Sec 5.2 show some improvements
> over two classical game-theoretic criteria, but the field of
> subset selection is now quite mature (e.g. https://arxiv.org/pdf/2006.15412.pdf), and comparing to just the two closest game-theoretic criteria
> doesn't seem to make a strong empirical case.
>
>
> **ANSWER**: Thanks for drawing connections with subset selection problems!   This is a misunderstanding regarding feature subset selection and valuation problems. And in this work we are studying general valuation problems in machine learning.
>
> We would like to clarify that feature valuation problems (also called feature attributions in interpretable machine learning) is different to feature subset selection: For valuation problems, one has to assign an importance score to each feature, while subset selection just needs to select a subset of features.
> Meanwhile, valuation problem is arguably harder than subset selection problem:
> Given valuation results, one can conduct subset selection according to the resulted importance score, but not vice versa.

---

> > ### Author Response · Authors · 2021-11-16
> > **Answers to Reviewer (part 2)**
> >
> >
> > ### Comment: cost of MCMC sampling
> > > The new criterion comes with the added cost of Monte Carlo sampling to approximate exponential sums over multiple rounds, but I'm not entirely sure if the experiments justify this added cost needed.
> >
> >
> > **ANSWER**: This is a misunderstanding regarding the exponential sum. It is
> > not an "added cost" of our method, in contrast, all game-theoretic valuation criteria in the paper have to handle this exponential cost.
> >
> > As discussed in the last paragraph of Sec. 4.2, all of them
> > suffer from this exponential cost. MCMC sampling
> > could help with approximating them.   We have also empirically verified
> > that Alg. 1  converges fast within 5 ~ 10 steps in Sec. 5.3.
> >
> >
> > ### Comment:
> > > The writing is accessible, but can be improved to better motivate
> > the subset/feature selection applications, especially for an ML audience,
> > and provide more elaborate intuition early on for the "decoupling" approximations that the classical criteria seek to compute.
> >
> >
> > **ANSWER**: Nice suggestion!  These valuation application come from various crucial ML
> > applications: feature valuation [Lundberg and Lee, 2017] comes from interpretation of ML models, data valuation comes from the emerging applications of data markets and collaborative machine learning [Ghorbani and Zou, 2019; Sim et al., 2020] and model valuation in ensembles [Rozemberczki and Sarkar,2021] comes from increasingly  more pretrained models published. One can see more details for these applications from the large
> > body of references in Section 3.
> >
> > We have  added more intuition in the revised version of our manuscript.
> >
> >
> > ### Other comments/questions:
> >
> > ### Comment:
> > > Alg 1: You refer to this algorithm as performing gradient ascent. To me it appears more like a fixed point iteration algorithm, and not necessarily performing a gradient-based ascent step on the ELBO objective. Please correct me if I'm missing something here.
> >
> > **ANSWER**: Thanks a lot for pointing this out! It is indeed more appropriate to be called a fixed point iteration. We changed it to be "fixed point" in the revised manu.
> >
> > A side note on this point is that:  The fixed point iteration actually
> > corresponds to gradient ascent with adaptive step size vector $\alpha^k$ as,
> >
> > $$\alpha_i^k = \frac{\sigma(\nabla_i f_{\text{mt}}^F(\mathbf{x}^{k})/T) - {x}_i^k }{\nabla_i f_{\text{mt}}^F(\mathbf{x}^{k})/T +\log[ {(1-{x}_i^k)}/{{x}_i^k} ]}$$
> >
> > ### Comment:
> > > Sec 5: Might be good to mention how the decoupling error is computed in all your experiments, given that it requires calculating (an approximation of) the Boltzmann distribution.
> >
> > **ANSWER**: As stated in Def. 1,  the decoupling error is defined as the divergence
> > between $q$ and $p$. In this work, it can be calculated as $\log \text{Z} - ELBO$. In order to rule out the influence of approximation errors when estimating their values,  we focus on small-sized problems where one can compute the exact values of them in a reasonable time.
> >
> > ### Comment:
> > > Sec 5.1: I didn't quite understand the role of the data clustering
> > in the experiment. On the x-axis in Fig 1, do you add clusters of
> > data points instead of individual data points?
> >
> > **ANSWER**:  Exactly as you said, we evaluate the values of data groups. The
> > data point setting is a special case of this setting with group size  set as
> > one.
> >
> > ### Comment: Evaluation metric for feature interpretations
> > > Sec 5.2: In Fig 2, you plot the predicted probabilities as you drop
> > features. Is this the average probability predicted by the model
> > across all test examples? Might be good to elaborate why this is a good evaluation metric to look at.
> >
> > **ANSWER**:   No, this is  for one single test sample, we log the data
> > point ID in the legend.
> > Additionally, we also report average results in Fig. 3 and more figures
> > in the appendix.
> >
> > Actually, this is just a common proxy metric in the area of interpretable
> > machine learning, where there is no golden standard to evaluate the
> > feature valuation results as far as we know.  This is also a common issue in the community of interpretable machine learning. We are also looking forward to golden standard evaluation metrics in this area.
> >
> >
> > ### Comment:
> > > Sec 1/Intro: I think your mention of a solution concept \phi(F) needs elaboration for audience not necessarily familiar with cooperative games. Similarly, in Sec 2, it wasn't initially clear to me how the
> > importance weights you write out in (2) and (3) relate to the solution concept \phi(F).
> >
> > **ANSWER**:  Good suggestion!
> > We give more explanation in the updated manuscript.

---

### Author Response · Authors · 2021-11-19
**Message to reviewers**

Dear Reviewers,

Thanks a lot to your valuable comments that help us to improve our work!  We are wondering whether your concerns have been addressed properly. We would be glad to answer any further questions you may have after reviewing the answers.


Best regards,

The authors.

---

### Author Response · Authors · 2021-11-21
**Main changes marked in blue color**

Dear Reviewers,


We have marked the main changes in the updated manuscript
in blue color, in order to better illustrate the refinement. They are mainly in Sections 1, 4.4, Appendices A, F.


Best,

The authors

---

### Decision · Program_Chairs · 2022-01-20

**Decision:**

Accept (Poster)

**Comment:**

This paper considers the valuation problem for a cooperative game, and shows that some classical metrics (e.g. Shapley value), can be considered as approximations to the maximum entropy.

Reviewers were generally very positive. They especially praised the novelty and writing quality, while having some concerns about the quality of the empirical results. The authors did an excellent job responding to the reviewers, and resolved their main concerns. A few quibbles remain, however, and while the manuscript is very good as-is, please consider the reviewer criticisms in creating an updated version.